# ECGN: A Cluster-Aware Approach to Graph Neural Networks for Imbalanced Classification

## Abstract

Classifying nodes in a graph is a common problem. The ideal classifier must adapt to any imbalances in the class distribution. It must also use information in the clustering structure of real-world graphs. Existing Graph Neural Networks (GNNs) have not addressed both problems together. We propose the Enhanced Cluster-aware Graph Network (*ECGN*), a novel method that addresses these issues by integrating cluster-specific training with synthetic node generation. Unlike traditional GNNs that apply the same node update process for all nodes, *ECGN* learns different aggregations for different clusters. We also use the clusters to generate new minority-class nodes in a way that helps clarify the inter-class decision boundary. By combining cluster-aware embeddings with a global integration step, *ECGN* enhances the quality of the resulting node embeddings. Our method works with any underlying GNN and any cluster generation technique. Experimental results show that *ECGN* consistently outperforms its closest competitors by up to 11% on some widely-studied benchmark datasets. The GitHub implementation for implementation and replication is publicly available on CodeLink.

## 1 Introduction

Graph Neural Networks (GNNs) have shown remarkable success in various tasks involving graph-structured data, including node classification (Kipf & Welling, 2016), link prediction (Zhang & Chen, 2018), and recommender systems (Ying et al., 2018). Indeed, GNNs have achieved state-of-the-art performance in many of these tasks. However, existing methods often expect all node classes and labels to be equally frequent. But in many real-world scenarios, node classes are imbalanced. For instance, most users on social network platforms are legitimate, but a small percentage are bots. This imbalance hurts the accuracy of bot detection (Mohammadrezaei et al., 2018). A similar challenge emerges when classifying websites by topics (Wang et al., 2020). A few topics are extremely popular, while most are rare. The popular topics (the majority classes) tend to dominate the loss function. Hence, the GNN focuses on these classes during training. This undermines the accuracy of the GNN for nodes of the minority classes. Hence, there is a need for GNN models capable of handling class-imbalanced node classification.

The class imbalance problem has been extensively studied in traditional machine learning. The solutions typically fall into three categories. *Data-level methods* balance the class distribution by over-sampling or under-sampling (Chawla et al., 2002; Kubat & Matwin, 1997). *Algorithm-level approaches* adjust the training process by using different misclassification penalties or prior probabilities for different classes (Ling & Sheng, 2008; Cui et al., 2019a). *Hybrid methods* combine both strategies to mitigate class imbalance (Batista et al., 2004a). However, all these methods assume independent and identically distributed data. By its very nature, the graph structure introduces dependencies between the nodes. Hence, such methods can yield suboptimal results when applied directly to graph datasets.

Compounding the class imbalance problem is the issue of *uniform node updates* in GNNs. Recall that GNNs update each node's embedding using information from the node's neighbors. The information exchange is mediated by trainable weight matrices. The same weights are uniformly applied to all nodes. However, such uniform node updates can cause two problems.

While standard GNNs use uniform update rules that aggregate information from neighboring nodes, they may not fully capture the rich local patterns and community-specific behaviors present in the graph. By incorporating cluster-aware updates, we aim to enhance the model's ability to learn these localized structures. Graphs have intricate substructures, such as clustered communities or hubs (Girvan & Newman, 2002). Nodes within the same cluster exhibit higher similarity and stronger dependencies than nodes from different clusters. Current GNNs can miss these nuanced local patterns and community-specific behaviors by treating all nodes identically. Second, since the weights are optimized over the entire graph, they can be biased towards the majority class. Hence, the node update process is sub-optimal for nodes from the minority class. These problems can lead to poor embedding quality and underperformance in classification.

While imbalanced classification and node clustering are two orthogonal problems, we argue that the interplay between them can be leveraged to address class imbalance more effectively. Specifically, clusters capture rich local structures and dependencies within the graph. By incorporating cluster-aware updates, we can more accurately learn the nuanced relationships between nodes within a cluster, mitigating the dominance of majority classes during training. Clustering provides a natural framework for focusing on minority-class nodes in their local context, enabling us to preserve their distinct patterns and improve their representation quality. Thus, combining these two aspects allows us to address both class imbalance and the limitations of uniform node updates simultaneously.

**Our Contributions:** Although some existing studies have addressed either label imbalance (Zhao et al., 2021a; Zhou & Gong, 2023a) or cluster-aware updates (Chiang et al., 2019), there is little work on tackling both issues simultaneously. We propose the *Enhanced Cluster-aware Graph Network (ECGN)* to bridge this gap.

*ECGN* operates through a three-phase process. In the **pre-training** phase, we train cluster-specific GNNs in parallel. These GNNs extract information from local structures in the graph while ensuring that all embeddings map to the same latent space. The **node generation** phase is a novel way to generate synthetic nodes for the minority class. In particular, the synthetic node representations incorporate cluster-specific features. Finally, the **global-integration** phase integrates the outputs of the previous stages into a cohesive set of node embeddings. These capture the global information across the entire graph. Our framework can be used with any existing GNN and is applicable whether or not the clusters are known a priori.

*ECGN* makes four significant contributions:

- **cluster-aware node updates,** that capture local cluster-specific information;
- **addressing label imbalance** via innovative synthetic cluster-aware node generation;
- **seamless local to global integration,** allowing the embeddings to learn from different scales; and
- **broad applicability,** by enabling any underlying GNN model to be used.

We verify the accuracy of *ECGN* on five benchmark datasets and show that we consistently outperform our closest competitors, with a **lift of up to** $11\%$ **in F1 score** on the widely studied Citeseer dataset. These results confirm the applicability of *ECGN* for a wide range of real-world applications.

## 2 RELATED WORKS

We discuss the related work on learning under class imbalance, and Graph Neural Networks.

### 2.1 CLASS IMBALANCE LEARNING

Class imbalance in representation learning is a well-established topic in the field of machine learning, having been extensively studied over the years (He & Garcia, 2009). The primary objective is to develop an unbiased classifier for a labeled dataset where the distribution is skewed, with majority classes having a substantially larger number of samples than minority classes. Notable contributions to this area include re-weighting and re-sampling techniques. Re-weighting methods modify the loss function by assigning greater importance to minority classes (Lin et al., 2017; Cui et al., 2019b), or by enlarging the margins for these classes (Cao et al., 2019; Liu et al., 2019; Menon et al., 2021).

On the other hand, re-sampling methods aim to balance the dataset by pre-processing the training samples, employing strategies like over-sampling the minority classes (Chawla et al., 2002), under-sampling the majority classes (Kubat & Matwin, 1997), or a combination of both (Batista et al., 2004b).

With advancements in neural networks, re-sampling strategies have evolved to not only include traditional sampling techniques but also to incorporate generative approaches. For instance, modern approaches augment minority class samples through generative methods (Liu et al., 2020), where techniques such as SMOTE (Chawla et al., 2002) generate new samples by interpolating between existing minority samples and their nearest neighbors. Additionally, other methods synthesize minority class samples by transferring knowledge from majority classes (Kim et al., 2020; Wang et al., 2021b). However, most of these existing methods are tailored to independent and identically distributed samples and are not directly applicable to graph-structured data, where the relational context between samples must be considered.

## 2.2 GRAPH NEURAL NETWORKS

Graph Neural Networks, first introduced in 2005 (Gori et al., 2005), have gained tremendous momentum in recent years with the advancements in deep learning, proving to be highly effective in processing non-Euclidean structured data. GNNs typically operate using a message-passing framework, where nodes iteratively gather information from their neighbors to learn low-dimensional embeddings that capture the graph's structural and feature information (Gilmer et al., 2017). These techniques are generally divided into two categories: spectral-based and spatial-based methods. Spectral-based methods exploit graph signal processing and leverage the graph Laplacian matrix to perform node filtering (Defferrard et al., 2016; Kipf & Welling, 2017; Bianchi et al., 2020), while spatial-based methods aggregate information directly from the local neighborhood of each node based on the graph topology (Veličković et al., 2018; Hamilton et al., 2017; You et al., 2019).

Addressing the challenge of class imbalance within GNNs has been an active area of research. Approaches such as GraphSMOTE (Zhao et al., 2021b) extend the popular SMOTE technique to the embedding space of GNNs by synthesizing new minority nodes while also generating additional edges, improving performance in imbalanced settings. Another approach, GraphENS (Park et al., 2022), generates synthetic minority node features by mixing existing nodes from other classes. In contrast, ClusterGCN (Chiang et al., 2019) leverages METIS-based graph partitioning to create subclusters and trains the GNN in an SGD-based framework to capture cluster-specific information. However, while ClusterGCN captures local structural information by operating on clusters, it still applies uniform node updates within each subcluster and across the entire graph, meaning the same update rules and aggregation functions are uniformly applied to all nodes without adapting to their unique local structures or roles within the graph. This uniformity prevents ClusterGCN from fully addressing the issue of uniform node updates during training, leading to a loss of fine-grained local-global patterns crucial for capturing nuanced relationships and dependencies in graph learning.

In addressing the class imbalance problem in graph data, our work builds upon and extends existing research that has explored various strategies for improving node classification performance under imbalanced conditions. Recent studies such Park et al. (2021) and Qian et al. (2022) propose novel methods for mitigating imbalance by modifying graph structures or introducing contrastive learning techniques. Wang et al. (2021a) introduced Distance-wise Prototypical GNNs, which focus on learning class prototypes in an imbalanced setting. Similarly, Song et al. (2022) with TAM, and Zeng et al. (2022) with Imgcl, highlight the importance of incorporating topological awareness and contrastive learning in handling imbalanced datasets. Moreover, Zhou & Gong (2023b) leverages data augmentation to improve minority-class representation. Other works include (Liu et al., 2023) which introduced an interesting topological augmentation framework for class imbalance, and (Li et al., 2023) which proposed a framework that synthesizes harder minor samples and incorporates a SemiMixup module to expand minority class decision boundaries without encroaching on neighboring class subspaces. While most of these methods focus on enhancing node embeddings, our approach distinguishes itself by integrating a cluster-aware framework, where we not only focus on adjusting the node updates but also generate synthetic minority-class nodes using a Cluster-Aware SMOTE technique. This results in a more robust decision boundary between classes, particularly when considering the underlying clustering structure in the graph. Our method, therefore, offers a complementary approach to the existing literature, enhancing both local cluster information and

global graph integration. We further demonstrate that ECGN outperforms these state-of-the-art techniques across several benchmark datasets, offering a more effective solution for imbalanced node classification.

# 3 PROPOSED ALGORITHM

We are given a graph $G = (V, E)$, where $V$ represents the set of nodes and $E$ denotes the set of edges. Each node $v_i \in V$ is associated with a feature vector $\mathbf{x}_i \in \mathbb{R}^d$, forming the node feature matrix $\mathbf{X} \in \mathbb{R}^{n \times d}$, where $n = |V|$ is the number of nodes and $d$ is the feature dimension. The graph structure is represented by the adjacency matrix $A \in \{0,1\}^{n \times n}$, where $A_{ij} = 1$ if there is an edge between nodes $v_i$ and $v_j$, and $A_{ij} = 0$ otherwise. Each node $v_i \in V$ belongs to a class $y_i \in \{Y_1, Y_2, \ldots, Y_c\}$, where $c$ is the number of classes. The class distribution can be imbalanced. The classes for a subset of the nodes are known, and our goal is to predict the classes for the remaining nodes.

Next, we discuss *ECGN*'s architecture and algorithm, and provide details of our novel node generation step.

## 3.1 ARCHITECTURE OF ECGN

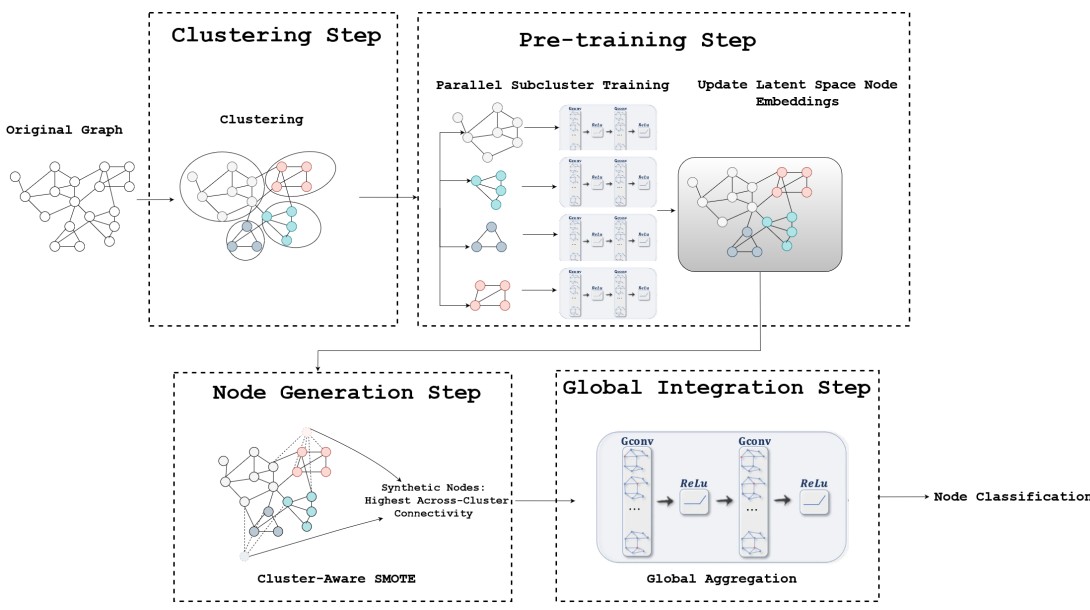

Figure 1: Working framework of ECGN architecture. We perform clustering on the nodes initially, train the sub-clusters independently in parallel, update the node embeddings to the original graph followed by Cluster-Aware SMOTE step and finally global integration.

The architecture of ECGN is presented in Figure 1. The algorithm begins by clustering the graph, unless the clusters are already known. Our method is flexible and can work with various clustering algorithms, though the choice of clustering algorithm does affect the results to some extent. We emphasize that the effectiveness of our method is not tied to any specific clustering approach, ensuring broad applicability across different scenarios. For example, fast algorithms such as Locality Sensitive Hashing (LSH) (Indyk & Motwani, 1998) or METIS (Karypis & Kumar, 1998) can be used for clustering (see Appendix A.4 for details).

Next comes the **pre-training** step. We first create a subgraph for each cluster. This subgraph includes only the nodes and edges within the cluster. Then, for each subgraph, we run a GNN

---

**Algorithm 1** ECGN for Node Classification

---

1: Initialize node features: $\mathbf{X} \in \mathbb{R}^{n \times d}$ and adjacency matrix: $\mathbf{A} \in \{0, 1\}^{n \times n}$.
2: Define the GNN architecture (network configuration) to be used for both pretraining and global integration.
3: **/* Clustering Step */**
4: Clusters $\mathcal{C} = \{C_1, C_2, \ldots, C_k\}$, either from prior knowledge, or obtained via *Locality Sensitive Hashing (LSH)* or *METIS partitioning* (Appendix A.4).
5: **/* Pre-training Step */**
6: Initialize GNN parameters $\theta$.
7: **for** each cluster $C_i \in \mathcal{C}$ **do**
8:     Construct subgraph $G_{C_i} = (V_{C_i}, E_{C_i})$ containing only nodes and edges from $C_i$.
9:     Train a GNN on $G_{C_i}$ with the objective of node classification within the cluster.
10:     Obtain embeddings $\mathbf{H}_{C_i} \leftarrow GNN(G_{C_i} \mid \theta)$ for the nodes in $C_i$.
11: **end for**
12: Combine embeddings: $\mathbf{H} \leftarrow \bigcup_{C_i \in \mathcal{C}} \mathbf{H}_{C_i}$.
13: **/* Synthetic Nodes Generation Step (Section 3.3) */**
14: Generate new embeddings for minority class nodes using Cluster-Aware SMOTE .
15: Create nodes corresponding to these embeddings, and add edges to them in the graph.
16: Call the new graph $G'$ with embeddings $\mathbf{H}'$
17: **/* Global Integration Step */**
18: $\mathbf{H}' \leftarrow$ convolution of $G'$ using $\mathbf{H}'$ as node features.
19: Perform node classification using the final embeddings $\mathbf{H}'$.

---

to generate embeddings for nodes in that cluster. All GNNs are run separately but share the same initialization. The result of this stage is a node embedding $\{\mathbf{h}_i\}$ for every node $i$ in the graph. Unlike embeddings from a global GNN, these embeddings focus on information from the local cluster of each node. Appendix A.2 provides a detailed description.

The next stage tackles the class imbalance problem. For this, we generate new nodes and edge with a new technique called **Cluster-Aware SMOTE**. This differs from standard SMOTE in several ways. First, our method operates on the latent space of the cluster-aware node embeddings $\mathbf{h}_i$ instead of the node features $\mathbf{x}_i$. Second, unlike SMOTE, our approach focuses on minority-class nodes that lie on cluster boundaries. The intuition is that in many real-world datasets, nodes from a given minority-class collect within one or a few clusters. Hence, the cluster boundaries are a proxy for inter-class decision boundaries. By generating nodes on the cluster borders, we can improve the decision boundary for node classification. Finally, after generating nodes with new embeddings, we link them to existing nodes. This step is needed for the following global aggregation step of *ECGN*. Details of Cluster-Aware SMOTE are presented in Section 3.3.

The last step is **global integration.** Here, we propagation global information throughout the nodes through graph convolution. The result is a set of node embeddings that combine information from local clusters as well as global patterns. Finally, these refined embeddings are used for node classification. Algorithm 1 provides the pseudo-code for *ECGN*.

## 3.2 PRE-TRAINING STEP

In the pre-training step, we aim to capture representative local features within each cluster by training cluster-specific GNN models independently. This process enhances the embeddings for each node by focusing on local structures relevant to its respective cluster.

We begin with the same initialization across all cluster-specific GraphSAGE models, ensuring consistency in the latent space. Each GraphSAGE model is trained with the same node classification objective but constrained to its assigned cluster subgraph. Specifically, for each cluster $C_i$, the GraphSAGE model learns node embeddings $\mathbf{h}_i^{(l+1)}$ at layer $l + 1$ as:

$$\mathbf{h}_i^{(l+1)} = \sigma \left( W_{\text{neigh}}^{(l)} \cdot \text{MEAN} \left( \{\mathbf{h}_j^{(l)} : j \in \mathcal{N}(i)\} \right) + W_{\text{self}}^{(l)} \cdot \mathbf{h}_i^{(l)} \right), \quad (1)$$

where:

---

- $\mathcal{N}(i)$ denotes the neighbors of node $i$ within cluster $C_i$,

- $W_{\text{neigh}}^{(l)}$ and $W_{\text{self}}^{(l)}$ are trainable weight matrices for aggregating neighbor and self-node embeddings at layer $l$, respectively,

- $\text{MEAN}(\cdot)$ computes the element-wise mean of the embeddings from neighbors $\{\mathbf{h}_j^{(l)} : j \in \mathcal{N}(i)\}$,

- $\sigma$ is a non-linear activation function (e.g., ReLU).

The objective for each cluster-specific GraphSAGE model is formulated as a node classification loss $\mathcal{L}_{\text{cluster}}^{(k)}$ for the $k$-th cluster:

$$\mathcal{L}_{\text{cluster}}^{(k)} = -\sum_{i \in C_k} \sum_{c=1}^{C} y_{ic} \log \hat{y}_{ic}, \tag{2}$$

where $y_{ic}$ is the true label of node $i$ for class $c$, and $\hat{y}_{ic}$ is the predicted probability for class $c$.

The combination of these cluster-specific embeddings results in enhanced node representations that capture unique, locally-relevant patterns. These embeddings are then passed to subsequent stages for integration with global information.

### 3.3 CLUSTER-AWARE SMOTE

Previous applications of SMOTE in graph contexts had several limitations. Synthetic nodes have been generated using graph features, but this ignores the link structure of the graph (Zhao et al., 2021b;a). Also, synthetic nodes were derived from minority-class seed nodes that were chosen randomly. If the seeds have poor connectivity, so do the synthetic nodes. Hence, the GNN's node updates may not efficiently convey information about the minority class.

Our proposed approach, named Cluster-Aware SMOTE, addresses these challenges by leveraging both intra- and inter-cluster connectivity information. Our method prioritizes minority-class nodes that lie on the borders of their clusters. The resulting synthetic nodes lead to a more accurate decision boundary between classes. Also, instead of using the node features, we use the cluster-aware node embeddings from the pre-training step. This ensures that both features and connectivity information is used in creating the synthetic nodes.

The synthetic node generation process involves the following steps:

1. **Identify Highly Connected Nodes:** For each minority node $v$ in class $Y_m$, we compute its connectivity to nodes in other clusters:

$$\text{connectivity}(v) = \sum_{\substack{u \in V \\ C(u) \neq C(v)}} A_{vu} \tag{3}$$

We select the top $k$ nodes with the highest connectivity scores as the seed nodes for generating synthetic samples.

2. **Nearest Neighbor Selection:** For each seed node $v$ from the minority class, we find its nearest neighbor $\text{nn}(v)$ within the same class in the embedding space:

$$\text{nn}(v) = \arg \min_{u \in Y_m, v \neq u} \|\mathbf{h}_v - \mathbf{h}_u\| \tag{4}$$

where $\mathbf{h}_i$ is the cluster-aware embedding for node $i$ from the pre-training step, and $\|\cdot\|$ denotes the Euclidean distance. For very large graphs, techniques like FAISS (Douze et al., 2024) can be used.

3. **Synthetic Node Generation:** We generate a synthetic node $v'$ by interpolating between the embeddings of $v$ and its nearest neighbor $\text{nn}(v)$:

$$\mathbf{x}_{v'} = (1 - \delta)\mathbf{x}_v + \delta \mathbf{x}_{\text{nn}(v)} \tag{5}$$

where $\delta$ is a random variable drawn from a uniform distribution in the range $[0, 1]$.

4. **Edge Preservation:** We add the synthetic node $v'$ to the graph, and add edges from $v'$ to all the neighbors of $v$. Thus, $v'$ inherits the edges of $v$. We also add a link from $v'$ to $v$. These steps ensure that the addition of the synthetic node $v'$ preserves the local graph topology. The updated adjacency matrix is as follows:

$$A_{v'i} = \begin{cases} A_{vi} & \text{if } i \neq v \\ 1 & \text{if } i = v \end{cases} \tag{6}$$

5. **Oversampling Control:** To control the number of generated nodes, we introduce a parameter $\alpha$. For each minority class $Y_m$, we generate $\alpha \cdot |Y_m|$ synthetic nodes. We restrict the value of $\alpha$ such that the number of synthetic nodes remains less than 50% of the majority class size. This ensures that the synthetic nodes do not degrade performance for the majority class.

By focusing on nodes with high inter-cluster connectivity and generating synthetic samples in the latent space, our approach improves the diversity of synthetic nodes and better captures the underlying graph structure. This not only helps in addressing class imbalance but also enhances the overall classification performance by providing more representative samples for the minority class.

## 4 EXPERIMENTS

We verify the accuracy of *ECGN* on five well-studied benchmark datasets. We first describe the datasets and the competing baselines. Then, we compare all algorithms on the node classification task. Finally, we show via ablation studies the need for the various steps of *ECGN*.

**Datasets:** We evaluate *ECGN* on several widely-used public datasets for the node classification task. All the details of datasets and baselines can be found in in Appendix A.1. Specifically, Table 2 shows the statistics and experimental setup for each dataset.

**Baselines:** We compared *ECGN* against several state of the art methods. These include GraphSAGE (with and without cluster information), cluster-aware GNNs such as ClusterGCN (Chiang et al., 2019), GNNs for imbalanced classification such as GraphSMOTE (Zhao et al., 2021b), recent state of the art models like GraphENS (Park et al., 2021) and TAM (Song et al., 2022) and various other reweighting and oversampling schemes. Appendices A.1 and A.5 provide more details about the baselines and their hyperparameters. All methods were tested on node classification tasks, and compared on the basis of their F1 scores. To ensure robust and reliable results, we averaged the F1-scores over four different random seeds.

**Direct Inference from Subclusters:** We evaluated the effect of bypassing global integration by directly inferring from subclusters without combining their representations into a global model. This experiment highlights the role of global integration in connecting local subcluster relationships with the global graph structure. Details of this experiment are provided in Appendix A.6.

**Weight Transfer Strategies:** We explored three strategies for transferring pre-trained GNN weights from subclusters to the global model: Average Weights, Largest Subcluster Weights, and Best Performing Subcluster Weights. These strategies were compared against *ECGN* without weight transfer to assess their influence on the global model's performance. For more details, see Appendix A.7.

**Sensitivity to Clustering Algorithms:** To evaluate the impact of different clustering methods, including METIS, LSH, and Random Clustering, we analyzed how they influence the performance of our method. This experiment was designed to assess the robustness of our approach to variations in graph partitioning techniques. Details are provided in Appendix A.9.

**Correlation between Clusters and Node Labels:** We analyzed the alignment between clusters and ground truth node labels (classes) by examining the distribution of class labels within clusters. This analysis provides insights into whether clustering processes naturally reflect the underlying label structure. See Appendix A.10 for a detailed explanation.

For *ECGN*, we clusters the graphs using METIS. We used 3 clusters for CORA and CITESEER, 7 for Amazon Computers, 40 for Reddit, and 20 for ogbn-arxiv. Section A.8 discusses how the number of clusters affect classification accuracy.

Table 1: *Results across different graph benchmark datasets:* Mean F1-Scores and Balanced Accuracies are reported along with standard deviations. *ECGN* with SMOTE outperforms other methods in both F1-Score and Balanced Accuracy. The lift is up to $10 - 12\%$ over the closest competitors (for Citeseer). *ECGN* with SMOTE is statistically significantly better than all competing methods at the $p < 0.1$ level except for the underlined rows.

(a) CORA

| Method | F1-Score | Balanced Accuracy |
|---|---|---|
| GraphSAGE | $0.655 \pm 0.04$ | $0.605 \pm 0.04$ |
| GraphSAGE (+ Cluster Features) | $0.680 \pm 0.04$ | $0.635 \pm 0.04$ |
| SMOTE | $0.690 \pm 0.03$ | $0.665 \pm 0.02$ |
| Re-Weighting | $0.670 \pm 0.02$ | $0.630 \pm 0.02$ |
| EN-Weighting | $0.680 \pm 0.04$ | $0.635 \pm 0.04$ |
| Over-Sampling | $0.645 \pm 0.03$ | $0.590 \pm 0.02$ |
| CB-Sampling | $0.710 \pm 0.02$ | $0.680 \pm 0.01$ |
| GraphSMOTE | $0.710 \pm 0.02$ | $0.680 \pm 0.01$ |
| GraphENS | $0.738 \pm 0.02$ | $0.712 \pm 0.03$ |
| TAM | $0.735 \pm 0.03$ | $0.720 \pm 0.03$ |
| ClusterGCN | $0.727 \pm 0.01$ | $0.690 \pm 0.03$ |
| Cluster-Aware SMOTE only | $0.700 \pm 0.02$ | $0.670 \pm 0.02$ |
| **ECGN (Without SMOTE)** | $\mathbf{0.732 \pm 0.03}$ | $\mathbf{0.710 \pm 0.03}$ |
| **ECGN (With SMOTE)** | $\mathbf{0.740 \pm 0.03}$ | $\mathbf{0.720 \pm 0.03}$ |

(b) CITESEER

| Method | F1-Score | Balanced Accuracy |
|---|---|---|
| GraphSAGE Baseline | $0.3625 \pm 0.04$ | $0.305 \pm 0.04$ |
| GraphSAGE (+ Cluster Features) | $0.3825 \pm 0.04$ | $0.325 \pm 0.04$ |
| SMOTE | $0.450 \pm 0.03$ | $0.430 \pm 0.03$ |
| Re-Weighting | $0.560 \pm 0.02$ | $0.530 \pm 0.02$ |
| EN-Weighting | $0.520 \pm 0.04$ | $0.500 \pm 0.04$ |
| Over-Sampling | $0.345 \pm 0.03$ | $0.300 \pm 0.02$ |
| CB-Sampling | $0.510 \pm 0.02$ | $0.490 \pm 0.02$ |
| GraphSMOTE | $0.590 \pm 0.02$ | $0.570 \pm 0.02$ |
| GraphENS | $0.630 \pm 0.02$ | $0.680 \pm 0.03$ |
| TAM | $0.625 \pm 0.03$ | $0.680 \pm 0.03$ |
| ClusterGCN | $0.580 \pm 0.03$ | $0.560 \pm 0.03$ |
| Cluster-Aware SMOTE only | $0.460 \pm 0.03$ | $0.430 \pm 0.03$ |
| **ECGN (Without SMOTE)** | $\mathbf{0.610 \pm 0.03}$ | $\mathbf{0.580 \pm 0.03}$ |
| **ECGN (With SMOTE)** | $\mathbf{0.650 \pm 0.03}$ | $\mathbf{0.620 \pm 0.03}$ |

(c) Reddit

| Method | F1-Score | Balanced Accuracy |
|---|---|---|
| GraphSAGE Baseline | $0.740 \pm 0.04$ | $0.700 \pm 0.04$ |
| GraphSAGE (+ Cluster Features) | $0.740 \pm 0.04$ | $0.700 \pm 0.04$ |
| SMOTE | $0.760 \pm 0.04$ | $0.730 \pm 0.04$ |
| Re-Weighting | $0.770 \pm 0.05$ | $0.740 \pm 0.05$ |
| EN-Weighting | $0.750 \pm 0.04$ | $0.720 \pm 0.04$ |
| Over-Sampling | $0.750 \pm 0.04$ | $0.720 \pm 0.04$ |
| CB-Sampling | $0.710 \pm 0.05$ | $0.690 \pm 0.05$ |
| GraphSMOTE | $0.770 \pm 0.05$ | $0.740 \pm 0.05$ |
| GraphENS | $0.780 \pm 0.05$ | $0.750 \pm 0.05$ |
| TAM | $0.770 \pm 0.05$ | $0.740 \pm 0.05$ |
| ClusterGCN | $0.760 \pm 0.04$ | $0.730 \pm 0.04$ |
| **ECGN (Without SMOTE)** | $\mathbf{0.770 \pm 0.05}$ | $\mathbf{0.740 \pm 0.05}$ |
| **ECGN (With SMOTE)** | $\mathbf{0.790 \pm 0.05}$ | $\mathbf{0.760 \pm 0.05}$ |

(d) ogbn-arxiv

| Method | F1-Score | Balanced Accuracy |
|---|---|---|
| GraphSAGE Baseline | $0.3675 \pm 0.04$ | $0.345 \pm 0.04$ |
| GraphSAGE (+ Cluster Features) | $0.3825 \pm 0.04$ | $0.355 \pm 0.04$ |
| SMOTE | $0.390 \pm 0.04$ | $0.370 \pm 0.04$ |
| Re-Weighting | $0.400 \pm 0.05$ | $0.380 \pm 0.05$ |
| EN-Weighting | $0.410 \pm 0.05$ | $0.390 \pm 0.05$ |
| Over-Sampling | $0.385 \pm 0.04$ | $0.365 \pm 0.04$ |
| CB-Sampling | $0.390 \pm 0.04$ | $0.370 \pm 0.04$ |
| GraphSMOTE | $0.410 \pm 0.04$ | $0.390 \pm 0.04$ |
| GraphENS | $0.440 \pm 0.05$ | $0.420 \pm 0.05$ |
| TAM | $0.430 \pm 0.05$ | $0.410 \pm 0.05$ |
| ClusterGCN | $0.430 \pm 0.04$ | $0.410 \pm 0.04$ |
| **ECGN (Without SMOTE)** | $\mathbf{0.442 \pm 0.04}$ | $\mathbf{0.430 \pm 0.05}$ |
| **ECGN (With SMOTE)** | $\mathbf{0.450 \pm 0.05}$ | $\mathbf{0.430 \pm 0.05}$ |

(e) Amazon Computers

| Method | F1-Score | Balanced Accuracy |
|---|---|---|
| GraphSAGE Baseline | $0.750 \pm 0.03$ | $0.720 \pm 0.03$ |
| GraphSAGE (+ Cluster Features) | $0.760 \pm 0.03$ | $0.730 \pm 0.03$ |
| SMOTE | $0.760 \pm 0.03$ | $0.730 \pm 0.03$ |
| Re-Weighting | $0.730 \pm 0.03$ | $0.700 \pm 0.03$ |
| EN-Weighting | $0.740 \pm 0.03$ | $0.720 \pm 0.03$ |
| Over-Sampling | $0.710 \pm 0.04$ | $0.670 \pm 0.04$ |
| CB-Sampling | $0.700 \pm 0.04$ | $0.680 \pm 0.04$ |
| GraphSMOTE | $0.760 \pm 0.03$ | $0.730 \pm 0.03$ |
| GraphENS | $0.770 \pm 0.04$ | $0.740 \pm 0.04$ |
| TAM | $0.760 \pm 0.04$ | $0.730 \pm 0.04$ |
| ClusterGCN | $0.740 \pm 0.03$ | $0.720 \pm 0.03$ |
| **ECGN (Without SMOTE)** | $\mathbf{0.767 \pm 0.04}$ | $\mathbf{0.740 \pm 0.04}$ |
| **ECGN (With SMOTE)** | $\mathbf{0.770 \pm 0.01}$ | $\mathbf{0.760 \pm 0.01}$ |

## 4.1 RESULTS

Table 1 shows the classification accuracy for all the datasets. We observe the following.

**ECGN consistently outperforms other models across all datasets.** The closest competitors are GraphENS, TAM , ClusterGCN and GraphSMOTE. However, *ECGN*'s F1-scores are higher by an average of nearly $5\%$. **ECGN outperforms its closest competitors by up to $11\%$** (e.g., on the Citeseer dataset).

**Cluster-aware node updates are necessary.** Consider two seemingly simple alternatives to the cluster-aware node updates of *ECGN*. One is to just provide the clusters as features to GraphSAGE.

The second is to just use the Cluster-Aware SMOTE without the pre-training step of *ECGN*. However, *ECGN* outperforms the former by $21\%$ on average, and the latter by $14\%$ on average.

**Cluster-Aware SMOTE improves classification accuracy.** The F1 score of *ECGN* using Cluster-Aware SMOTE is $3\%$ higher that *ECGN* without this step. For Citeseer, the difference increases to $6\%$. Thus, Cluster-Aware SMOTE adds value.

**Global integration is essential for performance.** Skipping global integration results in drastic F1-score reductions, with scores dropping to $0.26$ for Citeseer and $0.32$ for Cora. These results highlight that global integration is critical to effectively capture relationships between local subclusters and the global graph structure. Detailed results are in Appendix A.6.

**Direct training outperforms weight transfer strategies.** The weight transfer strategies (Average Weights, Largest Subcluster Weights, Best Performing Subcluster Weights) slightly improve performance, with the highest F1-score reaching $0.67$ on Cora. However, these approaches still underperform compared to direct global model training in *ECGN*, which achieves consistently higher accuracy. Refer to Appendix A.7 for the full comparison.

**ECGN is robust to different clustering strategies.** Our method demonstrates robust performance across various clustering algorithms, including METIS, LSH, and Random Clustering. This flexibility underscores the adaptability of *ECGN* to different graph partitioning techniques without significant degradation in performance. Details are provided in Appendix A.9.

**Clusters align meaningfully but not perfectly with class labels.** The analysis shows that clusters often align with specific class labels. For example, in Cora, Cluster 1 contains over $70\%$ of nodes from Class 2, while in Citeseer, Cluster 1 has a nearly equal mix of Classes 0 and 1. Similarly, in the Amazon Computers dataset, Cluster 7 has over $80\%$ of nodes from Class 4. However, not all clusters exhibit such strong alignment. These findings highlight that clustering processes capture meaningful patterns but do not always perfectly reflect class labels. See Appendix A.10 for a deeper analysis.

These experiments further validate the structure and robustness of ECGN, demonstrating its ability to generalize effectively across both local subcluster models and global models.

## 5    CONCLUSION AND LIMITATIONS

In this paper, we introduced the Enhanced Cluster-aware Graph Network (ECGN), a novel framework designed to address the challenges of class imbalance and subcluster-specific training in graph neural networks. By integrating cluster-specific updates, synthetic node generation, and a global integration step, *ECGN* demonstrates significant improvements in classification performance on imbalanced datasets. Our experimental results show that *ECGN* not only enhances the representation of minority classes but also maintains the structural integrity of the original graph, leading to more accurate and robust predictions. We also stated that modifying ECGN either by bypassing global tuning or integrating weight transfer learning hurts the performance. A detailed analysis of why ECGN works is provided in Appendix A.3 for more clarity.

However, there are limitations to our approach. The reliance on subcluster partitioning may introduce sensitivity to the quality of the clustering algorithm, and potentially impact the overall performance if the clusters are not well-formed. Additionally, the synthetic node generation process, while beneficial for handling imbalance, may introduce noise if not carefully managed, especially in graphs with highly complex structures. Future work will focus on refining these aspects for broader graph tasks like graph/link predictions, and evaluating *ECGN* on more diverse graph datasets to further validate its effectiveness.

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

# A APPENDIX

## A.1 DETAILS OF BASELINES AND EXPERIMENTAL DATASETS

In this section, we provide the details of the baselines implemented and experimental settings.

Table 2: Experimental settings to simulate imbalanced scenario for each datasets

| Dataset | # Classes | # Imbalanced Classes | Majority Class Samples | Minority Class Samples | Total Nodes | Total Edges | Validation Nodes | Testing Nodes |
|---|---|---|---|---|---|---|---|---|
| Cora | 7 | 3 | 200 | 20 | 2,708 | 5,429 | 2,050 | 1,426 |
| CiteSeer | 6 | 3 | 200 | 20 | 3,327 | 4,732 | 2,324 | 1,939 |
| Amazon Computers | 10 | 5 | 800 | 50 | 13,352 | 245,058 | 9,299 | 7,053 |
| Reddit | 40 | 10 | 1500 | 100 | 232,965 | 114,615,892 | 163,776 | 118,953 |
| ogbn-arxiv | 40 | 10 | 1500 | 100 | 169,343 | 1,166,851 | 118,540 | 86,348 |

- **Cluster-GCN:** A most popular GCN algorithm that is suitable for SGD-based training by exploiting the graph clustering structure based on METIS partitioning. (Chiang et al., 2019).

- **GraphSMOTE:** An oversampling method specifically designed for graphs that generates synthetic minority nodes by interpolating between existing nodes within the minority class (Zhao et al., 2021b).

- **Re-Weighting:** A classic cost-sensitive approach that adjusts the loss function with weights inversely proportional to the number of samples in each class (Japkowicz & Stephen, 2002).

- **EN-Weighting:** A variant of the re-weighting method, which assigns weights based on the Effective Number of samples in each class (Cui et al., 2019b).

- **Over-Sampling:** A traditional re-sampling method where minority nodes are repeatedly sampled until each minority class has the same number of samples as the majority classes.

- **CB-Sampling:** A re-sampling method inspired by (Butler, 1956), which first selects a class and then randomly samples a node from that class.

- **RU-Selection:** A baseline model that supplements the minority class by randomly selecting unlabeled nodes with pseudo-labels corresponding to the minority class until the class distribution is balanced.

- **SU-Selection:** An extension of RU-Selection that selects unlabeled nodes based on their similarity to the minority class, rather than random selection.

Here, we provide the details and explain the settings for the imbalanced scenario.

- **Cora Dataset:** Contains 2708 scientific publications categorized into 7 classes with 5429 links. We simulated a highly imbalanced scenario by sampling only 30% of the total samples available for the last 3 classes. Full-batch GD training was done, and number of METIS partition clusters were fixed to be 3. Synthetic nodes were added such that the minority class samples increases to 100 from 20 for each of the imbalanced class.

- **Citeseer Dataset:** Contains 3327 scientific publications classified into 6 categories with 4732 links. Full-batch GD training was done, and number of METIS partition clusters were fixed to be 3. Synthetic nodes were added such that the minority class samples increases to 100 from 20 for each of the imbalanced class.

- **Reddit Dataset:** Consists of posts made by users on the Reddit online discussion forum, categorized into 50 classes with over 230K nodes and 11M edges. The training was done with stochastic neighborhood sampling with batch size of 1024. The number of METIS partition clusters were fixed to be 40. Synthetic nodes were added such that the minority class samples increases to 400 from 50 for each of the imbalanced class.

- **Amazon Computers Dataset:** Contains 13,752 nodes categorized into 10 classes with 245,861 edges. Full-batch GD training was done, and the number of METIS partition clusters was fixed to 7. Synthetic nodes were added such that the minority class samples increases to 600 from 100 for each of the imbalanced class.

- **ogbn-arxiv Dataset:** Comprises 169,343 scientific publications from arXiv, categorized into 40 classes with 1,166,243 edges. To simulate an imbalanced scenario, we sampled only 100 nodes for the last 10 classes. The training was done with stochastic neighborhood sampling with batch size of 1024. The number of METIS partition clusters were fixed to be 20. Synthetic nodes were added such that the minority class samples increases to 600 from 100 for each of the imbalanced classes.

## A.2 ORIGINAL GRAPHSAGE VS. SUBCLUSTERED GRAPHSAGE

**Original GraphSAGE** GraphSAGE (Hamilton et al., 2017) generates node embeddings by aggregating features from a node's local neighborhood. Given a graph $G = (V, E)$ with $N = |V|$ nodes and initial node features $\mathbf{X} \in \mathbb{R}^{N \times F}$, the embedding of node $v$ at layer $k$ is updated as:

1. **Neighborhood Aggregation:**

$$\mathbf{h}_{\mathcal{N}(v)}^{(k)} = \text{AGGREGATE}^{(k)} \left( \left\{ \mathbf{h}_u^{(k-1)} \mid u \in \mathcal{N}(v) \right\} \right), \tag{7}$$

where $\mathcal{N}(v)$ is the set of neighbors of node $v$, and $\mathbf{h}_u^{(k-1)}$ is the embedding from the previous layer.

2. **Node Embedding Update:**

$$\mathbf{h}_v^{(k)} = \sigma \left( \mathbf{W}^{(k)} \cdot \text{CONCAT} \left( \mathbf{h}_v^{(k-1)}, \mathbf{h}_{\mathcal{N}(v)}^{(k)} \right) \right), \tag{8}$$

where $\mathbf{W}^{(k)}$ is the weight matrix, $\sigma$ is an activation function, and $\mathbf{h}_v^{(0)} = \mathbf{x}_v$.

This process is repeated for $K$ layers to capture $K$-hop neighborhood information. The final embeddings $\mathbf{h}_v^{(K)}$ are used for tasks like node classification.

**Subcluster-Based GraphSAGE**   In *ECGN*, we enhance GraphSAGE by incorporating cluster-specific information:

1. **Graph Partitioning:** Divide $G$ into $M$ disjoint subclusters $\{G_1, G_2, \ldots, G_M\}$, with corresponding feature matrices $\mathbf{X}_i$.

2. **Localized Learning:** For each subcluster $G_i$, perform GraphSAGE focusing only on Cluster-Aware edges:

$$\mathbf{h}_v^{(k)} = \sigma \left( \mathbf{W}^{(k)} \cdot \text{CONCAT} \left( \mathbf{h}_v^{(k-1)}, \text{AGGREGATE}_i^{(k)} \left( \left\{ \mathbf{h}_u^{(k-1)} \mid u \in \mathcal{N}_i(v) \right\} \right) \right) \right), \tag{9}$$

where $\mathcal{N}_i(v)$ denotes Cluster-Aware neighbors.

3. **Embedding Compilation:** Combine embeddings from all subclusters:

$$\mathbf{H}^{(K)} = \begin{pmatrix} \mathbf{H}_1^{(K)} \\ \mathbf{H}_2^{(K)} \\ \vdots \\ \mathbf{H}_M^{(K)} \end{pmatrix}. \tag{10}$$

4. **Global Integration:** Perform an additional GraphSAGE layer over $G$ to integrate global information:

$$\mathbf{h}_v^{(\text{final})} = \sigma \left( \mathbf{W}^{(K+1)} \cdot \text{CONCAT} \left( \mathbf{h}_v^{(K)}, \text{AGGREGATE}^{(K+1)} \left( \left\{ \mathbf{h}_u^{(K)} \mid u \in \mathcal{N}(v) \right\} \right) \right) \right). \tag{11}$$

**Key Advantages**

- **Enhanced Local Patterns:** Captures fine-grained structures within clusters.
- **Computational Efficiency:** Allows parallel processing of subclusters.
- **Global Coherence:** Global aggregation integrates inter-cluster relationships.
- **Improved Handling of Imbalance:** Clustering aids in addressing class imbalance by focusing on underrepresented nodes within clusters.

By combining localized learning with global integration, the subcluster-based approach in *ECGN* effectively captures both local and global graph structures, leading to improved performance in node classification tasks.

### A.3 Visualizing the Clustered Communities and Analyzing the Sub-Clustered Approach

In this section, we visualize the clustered communities within the Cora and Citeseer datasets and we try to provide a theoretical explanation of why our sub-clustered approach is effective. We selected these datasets due to their manageable size and well-documented structure, which makes them ideal candidates for visual analysis. We divided the datasets into three clusters using the METIS algorithm and present the visualizations below.

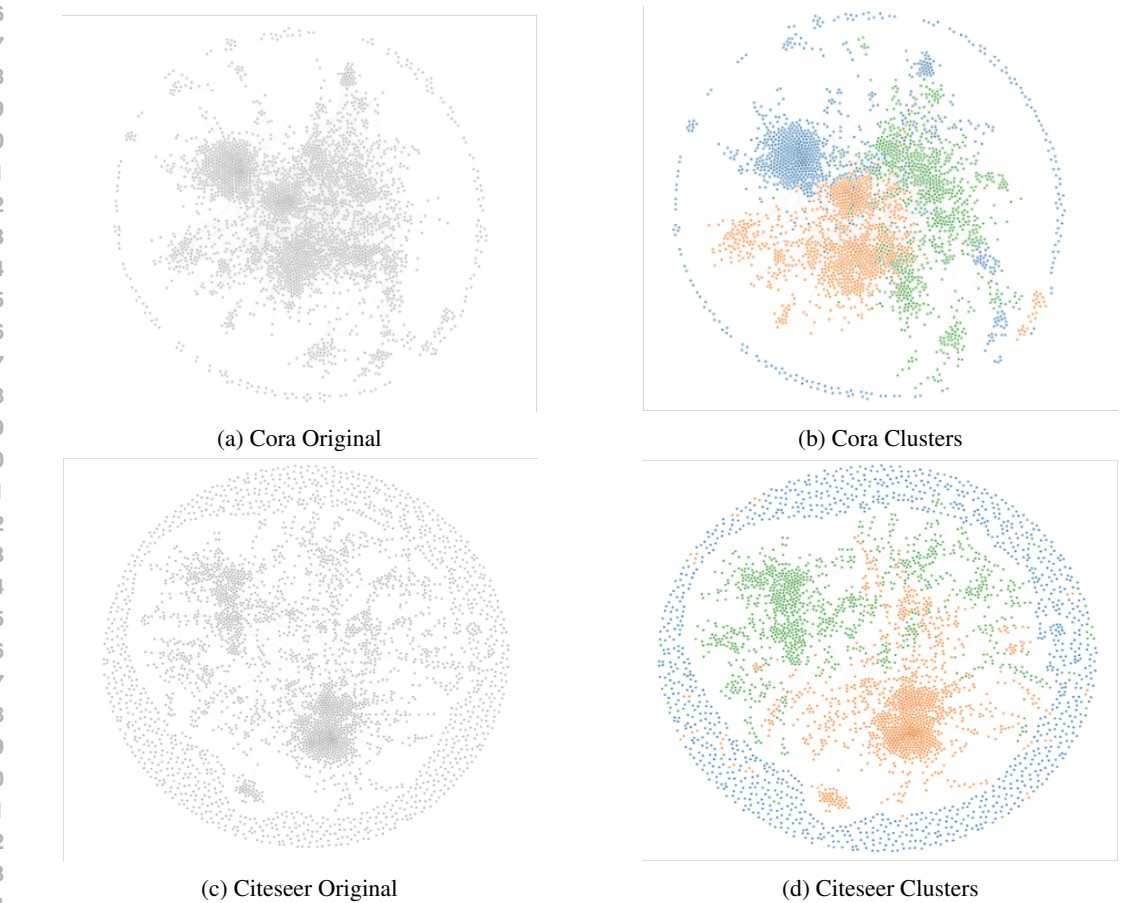

(a) Cora Original

(b) Cora Clusters

(c) Citeseer Original

(d) Citeseer Clusters

Figure 2: Visualizations of the original and clustered versions of the Cora and Citeseer datasets. The left column shows the original datasets, while the right column shows the datasets divided into three clusters using METIS clustering.

The figures in Figure 2 depict both the original and clustered versions of the Cora and Citeseer datasets. The original graphs (Figure 2a and Figure 2c) exhibit dense connectivity, which often leads to an entangled representation where the underlying community structure is not immediately apparent. By applying the METIS algorithm, we break down these dense graphs into distinct clusters (Figure 2b and Figure 2d), revealing the internal structure of the communities.

In this section, we try to provide a theoretical explanation of why our sub-clustered approach is effective.

**Why does the Sub-Clustered Approach Work?**   The effectiveness of the sub-clustered approach can be theoretically explained through the following principles:

**1. Capturing Localized Patterns**   When the graph is clustered into $k$ subgraphs using METIS, we obtain subgraphs $G_1, G_2, \ldots, G_k$ such that:

$$G_i = (V_i, E_i), \quad \text{where} \quad V_i \subseteq V \quad \text{and} \quad E_i \subseteq E$$

The feature matrix for each subgraph is $\mathbf{X}_i \in \mathbb{R}^{n_i \times d}$, where $n_i = |V_i|$ is the number of nodes in subgraph $G_i$.

By training on these subgraphs independently, the model learns localized patterns within each $G_i$, which are typically more homogeneous and easier to capture than the global patterns in $G$. The local

loss function for each subgraph can be expressed as:

$$\mathcal{L}_i = \frac{1}{n_i} \sum_{v_j \in V_i} \mathcal{L}(f(\mathbf{x}_j), y_j)$$

where $f(\mathbf{x}_j)$ is the model's prediction for node $v_j$, and $y_j$ is the true label. Training on localized loss functions $\mathcal{L}_i$ allows the model to optimize performance within each cluster before aggregating the knowledge during global integration.

**2. Reducing Computational Complexity**    The computational complexity of training a GNN on a large graph $G$ is often dominated by the cost of message passing and aggregation across the entire graph. However, by decomposing $G$ into smaller subgraphs $G_1, G_2, \ldots, G_k$, and being able to train them parallel independently, the computational cost is significantly reduced.

The overall complexity can be approximated as:

$$\text{Total Complexity} \approx \sum_{i=1}^{k} \mathcal{O}(|E_i|)$$

where $|E_i|$ is the number of edges in subgraph $G_i$. Since each $|E_i|$ is smaller than $|E|$ (the total number of edges in the original graph), the sub-clustered approach leads to more efficient training.

**3. Addressing Imbalanced Data**    In imbalanced graphs, certain classes of nodes may be under-represented, making it difficult for the model to learn their characteristics. By isolating these nodes within sub-clusters, the model can pay more focused attention to the minority classes.

Let $C$ be the set of classes in the graph, with $|C_{\min}|$ and $|C_{\maj}|$ representing the number of nodes in the minority and majority classes, respectively. After clustering, the number of minority nodes in a subgraph $G_i$ can be denoted as $|C_{\min,i}|$. The training process can now focus on balancing the loss contributions:

$$\mathcal{L}_i^{\text{balance}} = \frac{1}{|C_{\min,i}|} \sum_{v_j \in C_{\min,i}} \mathcal{L}(f(\mathbf{x}_j), y_j) + \frac{1}{|C_{\maj,i}|} \sum_{v_j \in C_{\maj,i}} \mathcal{L}(f(\mathbf{x}_j), y_j)$$

This ensures that the minority class nodes have a more significant influence on the model's learning process within each cluster.

**4. Global Structure Integration**    After the initial training on sub-clusters, the model undergoes global aggregation on the global graph $G$. This step integrates the knowledge learned from each sub-cluster and ensures that node representations are coherent across the entire graph. The global integration process can be represented as:

$$\mathcal{L}_{\text{global}} = \frac{1}{n} \sum_{v_j \in V} \mathcal{L}(f(\mathbf{x}_j), y_j)$$

This global loss function aligns the local representations and improves the overall performance of the model.

By training on these clusters and then performing the global integration, the model leverages both localized knowledge and global context, resulting in more accurate and generalizable node representations. The sub-clustered approach mitigates the risk of overfitting to dominant structures and promotes a more balanced and comprehensive understanding of the graph.

A.4    CLUSTERING AND COMMUNITY DETECTION TECHNIQUES

A.4.1    LOCALITY-SENSITIVE HASHING CLUSTERING (FEATURE BASED CLUSTERING)

We present an standard version of LSH clustering algorithm to efficiently handle large-scale datasets with millions of nodes. The algorithm uses sparse random projections to group similar feature vectors into clusters.

**Algorithm Description**

Given a feature matrix $\mathbf{X} \in \mathbb{R}^{n \times d}$, where $n$ is the number of samples and $d$ is the number of features, the goal is to cluster the samples based on their similarity. The optimized algorithm proceeds as follows:

1. **Hash Table Creation:** We create $T$ hash tables, each using $P$ random projections. Each hash table is represented by a sparse random projection matrix $\mathbf{R} \in \mathbb{R}^{d \times P}$.

$$\mathbf{R}_t \sim \text{SparseRandomProjection}(d, P) \quad \text{for} \quad t = 1, 2, \ldots, T$$

2. **Hashing Feature Vectors:** Each feature vector $\mathbf{x}_i \in \mathbb{R}^d$ is projected into a lower-dimensional space using the hash tables. The projection is followed by taking the sign of the resulting vector to create hash keys.

$$\mathbf{h}_i^t = \text{sign}(\mathbf{x}_i \mathbf{R}_t) \quad \text{for} \quad i = 1, 2, \ldots, n \quad \text{and} \quad t = 1, 2, \ldots, T$$

The sign function is applied element-wise, resulting in a hash key $\mathbf{h}_i^t \in \{-1, 1\}^P$.

3. **Bucket Assignment:** Each hash key is used to group feature vectors into buckets. A bucket $\mathcal{B}_t(\mathbf{h})$ contains all vectors that share the same hash key $\mathbf{h}$ for the $t$-th hash table.

$$\mathcal{B}_t(\mathbf{h}) = \{i \mid \mathbf{h}_i^t = \mathbf{h}\}$$

4. **Merging Buckets:** We merge buckets from all hash tables into preliminary clusters. Each node is assigned to a cluster based on its initial bucket assignments, ensuring unique assignments.

$$\text{clusters}[i] = \text{cluster\_id} \quad \text{if} \quad i \in \mathcal{B}_t(\mathbf{h}) \quad \forall t$$

5. **Cluster Refinement:** Each preliminary cluster is refined by computing the centroid of its feature vectors and using cosine similarity with a threshold to ensure nodes belong to the most similar cluster.

$$\text{similarity}(\mathbf{x}_i, \mathcal{C}) = \frac{\mathbf{x}_i \cdot \mathbf{c}_\mathcal{C}}{\|\mathbf{x}_i\|\|\mathbf{c}_\mathcal{C}\|} \quad \text{where} \quad \mathbf{c}_\mathcal{C} = \frac{1}{|\mathcal{C}|} \sum_{\mathbf{x}_j \in \mathcal{C}} \mathbf{x}_j$$

Each node $i$ is assigned to cluster $\mathcal{C}^*$ if $\text{similarity}(\mathbf{x}_i, \mathcal{C}^*) > 0.5$.

6. **Final Cluster Formation:** The final clusters are formed by ensuring each node belongs to one and only one cluster.

$$\mathcal{C}_k \rightarrow \mathbf{C}_k \quad \text{where} \quad \mathbf{C}_k \in \mathbb{R}^{|\mathcal{C}_k| \times d}$$

The algorithm ensures efficient clustering of high-dimensional data by leveraging the properties of locality-sensitive hashing and sparse random projections. The resulting clusters can be used in subsequent tasks such as classification, anomaly detection, and data summarization.

### A.4.2 METIS PARTITIONING (STRUCTURE BASED CLUSTERING)

*METIS partitioning* is a graph partitioning technique designed to divide a graph into smaller, roughly equal-sized subgraphs while minimizing the edge cuts between them. The primary objective is to balance the load across subgraphs and reduce the communication volume in parallel computing environments.

Given a graph $G = (V, E)$ with vertices $V$ and edges $E$, the goal is to partition $G$ into $k$ subgraphs $G_1, G_2, \ldots, G_k$ such that:

1. The size of each subgraph is approximately equal, i.e., $|V_i| \approx \frac{|V|}{k}$ for $i = 1, 2, \ldots, k$. 2. The number of edges cut, denoted as $\text{cut}(G)$, is minimized. This is mathematically represented as minimizing the sum of weights of edges that have endpoints in different subgraphs:

$$\text{cut}(G) = \sum_{\substack{(u,v) \in E \\ u \in G_i, v \in G_j \\ i \neq j}} w(u, v)$$

where $w(u, v)$ is the weight of the edge between nodes $u$ and $v$.

METIS employs a multilevel approach, which involves three main phases:

1. **Coarsening Phase**: The graph is iteratively coarsened by collapsing vertices and edges to form a series of progressively smaller graphs.
2. **Partitioning Phase**: A partitioning algorithm, often a variant of the *Kernighan-Lin* or *Fiduccia-Mattheyses* heuristic, is applied to the smallest graph to obtain an initial partition.
3. **Uncoarsening Phase**: The initial partition is projected back through the series of intermediate graphs, refining the partition at each level to improve the quality of the final partition.

This multilevel approach ensures that the partitioning process is both efficient and effective in producing high-quality partitions with balanced subgraph sizes and minimal edge cuts.

## A.5 Experimental settings for Baseline experiments

In our experiments, we use METIS partitioning to create subclusters for all the datasets. The experiments were configured with consistent training hyperparameters across datasets, including an initial learning rate of 0.01 for most datasets (with Reddit using 0.001), and the Adam optimizer. Each experiment ran for up to 1500 epochs, with early stopping after 40 steps if the validation performance did not improve. A batch size of 128 was used for Cora and Citeseer, while larger datasets such as Reddit, AmazonComputer, and ogbn_arxiv used batch sizes of 1024 or 2048 to accommodate their size. For model architecture, we adopted 2-layer Graph-SAGE with a layer dimension of 128 for most datasets, though Reddit employed a smaller 64-dimensional GNN with 1 layer. We used the 'mean' aggregator for message passing and allowed for dynamic learning rates across layers. Full-batch training was used for Cora, Citeseer, and ogbn_arxiv, whereas AmazonComputer and Reddit were trained with neighborhood sampling (as documented in `https://docs.dgl.ai/en/0.8.x/guide/minibatch-node.html#guide-minibatch-node-classification-sampler`) with 4-layer deep neighborhood samples with sizes [4,4,4,4] due to their size. Additionally, datasets were clustered, with the number of clusters ranging from 3 (Cora, Citeseer), 7(AmazonComputer), 20(obgn-arxiv) to 40 (Reddit).

## A.6 Direct Inference from Subclusters: Why Do We Need Global Integration?

In this section, we explore the impact of bypassing the global integration step and directly inferring from subclusters. This approach leverages local structures within each subcluster but neglects the global graph structure. To evaluate this, we conducted experiments on the Cora and Citeseer datasets, which offer manageable complexity for detailed analysis.

Table 3: Performance results when directly inferring from subclusters without global integration.

| Dataset | Num Clusters | F1-Score Without Global Integration | Best *ECGN* F1-Score |
|---------|--------------|-------------------------------------|----------------------|
| Citeseer | 3 | 0.26 | 0.65 |
| Cora | 3 | 0.32 | 0.74 |

Table 3 shows that skipping global integration leads to significantly lower F1-scores: 0.26 for Citeseer and 0.32 for Cora. This highlights the importance of integrating the global graph after subcluster training. Without it, the model learns *Cluster-Aware* relations but fails to generalize and learn *inter-cluster* relations, causing lower performance.

In conclusion, the global integration step is crucial as it bridges the gap between local subcluster structures and the overarching global graph. It ensures that the final node embeddings are both locally accurate and globally consistent, leading to better performance, as evidenced by the increased F1-scores after global integration.

## A.7 Reusing GNN Weights from Pre-training in Global Integration

We can think of *ECGN* as a transfer learning approach. In the pre-training step, we learn separately from each cluster. Then, we transfer the learnt embeddings to the global integration step.

This ensures a balance between local and global structures in the graph, which yields the strong performance of *ECGN*.

Extending this idea, we can ask: what if we transferred the GNN model weights from pre-training alongside the node embeddings? To explore this, we experimented with three different strategies for transferring weights:

1. **Average Weights**: Initialize the weights of the global GNN with the averaged weights of all subcluster GNNs.

2. **Largest Subcluster Weights**: GNN weights from the largest subcluster are transferred to the global model.

3. **Best Performing Subcluster Weights**: Weights from the best-performing subcluster are transferred to the global model.

Table 4: Performance results when transferring weights along with feature representations across different strategies.

| Dataset | Weight Transfer Strategy | F1-Score with Weight Transfer | Best *ECGN* F1-Score |
|---|---|---|---|
| **Citeseer** | Average | 0.51 | 0.65 |
| | Largest | 0.65 | 0.65 |
| | Best | 0.66 | 0.65 |
| **Cora** | Average | 0.52 | 0.74 |
| | Largest | 0.66 | 0.74 |
| | Best | 0.67 | 0.74 |
| **Amazon Computers** | Average | 0.54 | 0.78 |
| | Largest | 0.68 | 0.78 |
| | Best | 0.69 | 0.78 |

The results in Table 4 show that transferring pre-trained weights alongside the node embeddings yields mixed outcomes. The *Average Weights* strategy consistently performed the worst, likely because averaging diluted the unique structural information from each subcluster. The *Largest Subcluster Weights* and *Best Performing Subcluster Weights* improved performance over averaging but did not outperform *ECGN* without weight transfer. Overall, we find that the best performance comes from ignoring the pre-trained GNN weights in the global integration. This is what *ECGN* does.

We believe the reason for the above results lies in the delicate balancing act between learning from the graph's local structure and its global context. The introduction of pre-trained weights $\mathbf{W}$ into the global model appears to disrupt this balance. The pre-trained weights are perhaps too tailored to specific clusters. So, they may not generalize well when applied to the entire graph.

In conclusion, weight transfer offers no significant advantage and may reduce performance. The success of *ECGN* lies in combining local embeddings through global integration, capturing both local and global structures without transferring subcluster-specific weights.

## A.8 SELECTING THE NUMBER OF CLUSTERS

The number of clusters affects both the effectiveness of the cluster-aware embeddings and *ECGN*'s computational efficiency. The key challenge lies in balancing the capture of fine-grained local patterns with the retention of important global structures.

In our experiments, we observed that the model's performance is relatively robust to the number of clusters within a reasonable range. As shown in Figure 3, the F1-Score varies with the number of clusters across different datasets. For datasets like Cora and Citeseer, using **3 to 5 clusters** yielded comparable results, while extreme values—either too low or too high—led to decreased performance. This suggests that while the number of clusters is important, the model can tolerate variations without significant loss of effectiveness.

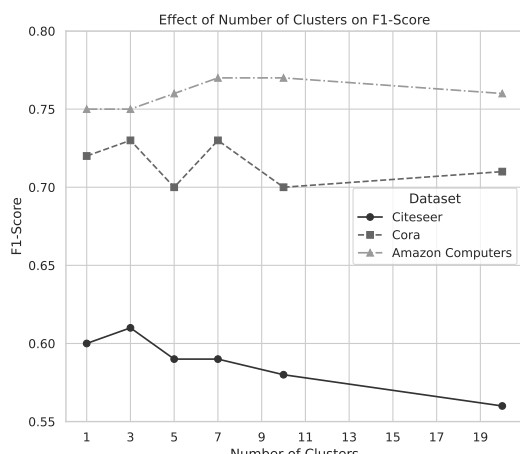

Figure 3: Effect of Number of Clusters on F1-Score for Various Datasets

## A.9 Sensitivity to Clustering Algorithms

We evaluated the impact of different clustering algorithms on our method's performance by conducting experiments using METIS (a structure-based clustering algorithm), Locality-Sensitive Hashing (LSH, a feature-based clustering approach), and Random Clustering. These experiments were performed on the Cora and Citeseer datasets to assess how varying clustering strategies influence the overall effectiveness of our approach.

Table 5: Impact of Different Clustering Algorithms on Performance (F1-Score)

| Clustering Algorithm | Cora | Citeseer |
|---|---|---|
| METIS | $0.740 \pm 0.03$ | $0.650 \pm 0.03$ |
| LSH | $0.729 \pm 0.02$ | $0.639 \pm 0.02$ |
| Random | $0.690 \pm 0.05$ | $0.610 \pm 0.03$ |

As anticipated, METIS outperforms both LSH and Random Clustering on both datasets. METIS leverages the underlying graph structure to create clusters that reflect the inherent connectivity of the data, resulting in higher F1-scores. LSH, being a feature-based method, also provides meaningful clusters, though with slightly lower performance compared to METIS. Interestingly, Random Clustering achieves competitive results despite lacking semantic or structural coherence. This highlights the robustness of our method, which effectively utilizes localized training within clusters to mitigate class imbalance issues. By grouping and balancing nodes locally, our approach reduces the dominance of majority classes, thereby enhancing overall performance even when the clustering quality is suboptimal.

The performance of Random Clustering, although lower than METIS and LSH, remains surprisingly competitive. This can be attributed to our method's ability to perform localized training within each cluster, ensuring that each smaller group of nodes is more balanced. This reduces the dominance of majority classes within each cluster, allowing the model to learn more effectively. Therefore, the method demonstrates significant robustness, maintaining effectiveness even when the clustering lacks inherent semantic or structural coherence.

In scenarios where clustering aligns perfectly with class labels, the performance of our method could potentially see further improvements due to reduced intra-cluster variance and better separation between classes. However, it is important to note that our method does not rely on such perfect alignment and achieves strong performance even with imperfect clustering, underscoring its flexibility and adaptability to various clustering qualities.

Overall, the ablation study and subsequent analysis demonstrate that our method is both robust and flexible, capable of maintaining high performance across different clustering algorithms. This

adaptability is a key strength, allowing our approach to be effectively applied in various contexts and with different types of data clustering strategies.

### A.10   CORRELATION BETWEEN CLUSTERS AND NODE LABELS(CLASSES)

To better understand the relationship between clusters and node labels (classes), we conducted an analysis examining how nodes with specific labels are distributed across clusters. Clustering algorithms, such as Locality-Sensitive Hashing (LSH) and METIS, group nodes based on their features or structural properties. However, these clusters are formed independently of the node labels, making it important to investigate the alignment (or lack thereof) between clusters and classes.

This analysis was performed on three datasets: Cora, Citeseer, and Amazon Computers. Figures 4, 5, and 6 illustrate the distribution of node classes within clusters for each dataset. Each bar plot in these figures shows the percentage of nodes from different classes within a specific cluster.

**Cora Dataset:** In the Cora dataset (Figure 4), certain clusters exhibit a strong alignment with specific classes. For instance, Cluster 1 contains over 70% of nodes from Class 2, suggesting that the clustering algorithm successfully grouped nodes with similar features from this class. However, other clusters, such as Cluster 3, display a more diverse mix of classes, indicating that nodes with overlapping features across classes can be grouped together.

**Citeseer Dataset:** The Citeseer dataset (Figure 5) reveals a more fragmented relationship between clusters and classes. For example, Cluster 2 is dominated by Class 2, while Cluster 1 shows an almost equal mix of Classes 0 and 1. This variation reflects both the dataset's inherent label imbalance and the clustering algorithm's sensitivity to feature similarities. Such patterns emphasize that clusters may not always align perfectly with specific classes.

**Amazon Computers Dataset:** The Amazon Computers dataset, with its larger number of classes, exhibits both highly concentrated and diverse clusters (Figure 6). For example, Cluster 7 is dominated by Class 4, with over 80% of its nodes belonging to this class. In contrast, Cluster 2 contains nodes from a wider range of classes, showcasing the clustering algorithm's adaptability in handling datasets with complex structures and larger numbers of classes.

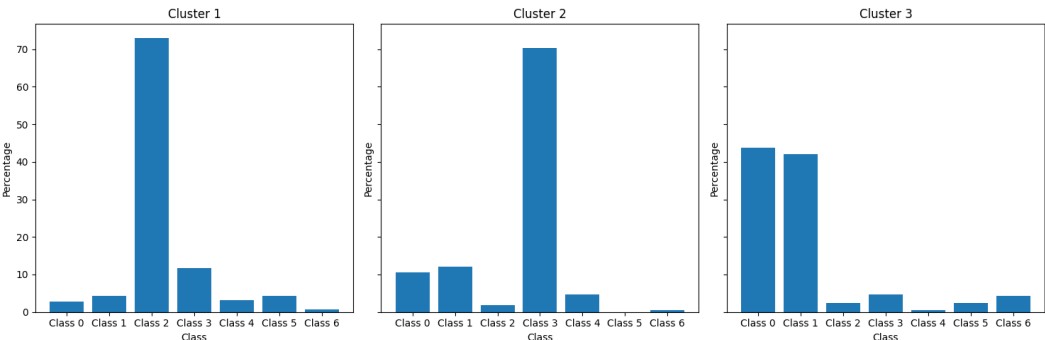

Figure 4: Class distributions across clusters for the Cora dataset.

**Key Observations:**

- Clusters often show a concentration of nodes from one or two dominant classes, indicating that clustering algorithms effectively group nodes with similar features or structural relationships.
- In some cases, nodes from different classes are grouped together, especially when features overlap or structural relationships between nodes span across classes.
- Datasets with more classes and higher complexity, such as Amazon Computers, display a mix of highly concentrated and diverse clusters, reflecting the adaptability of the clustering approach to different data characteristics.

**Implications for Methodology:** This experiment demonstrates that while clusters and classes may align in some cases, this alignment is not deterministic. The observed correlations indicate that

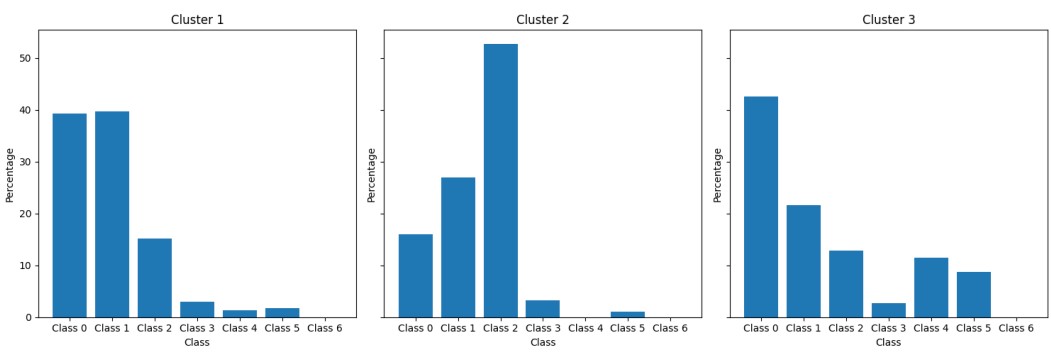

Figure 5: Class distributions across clusters for the Citeseer dataset.

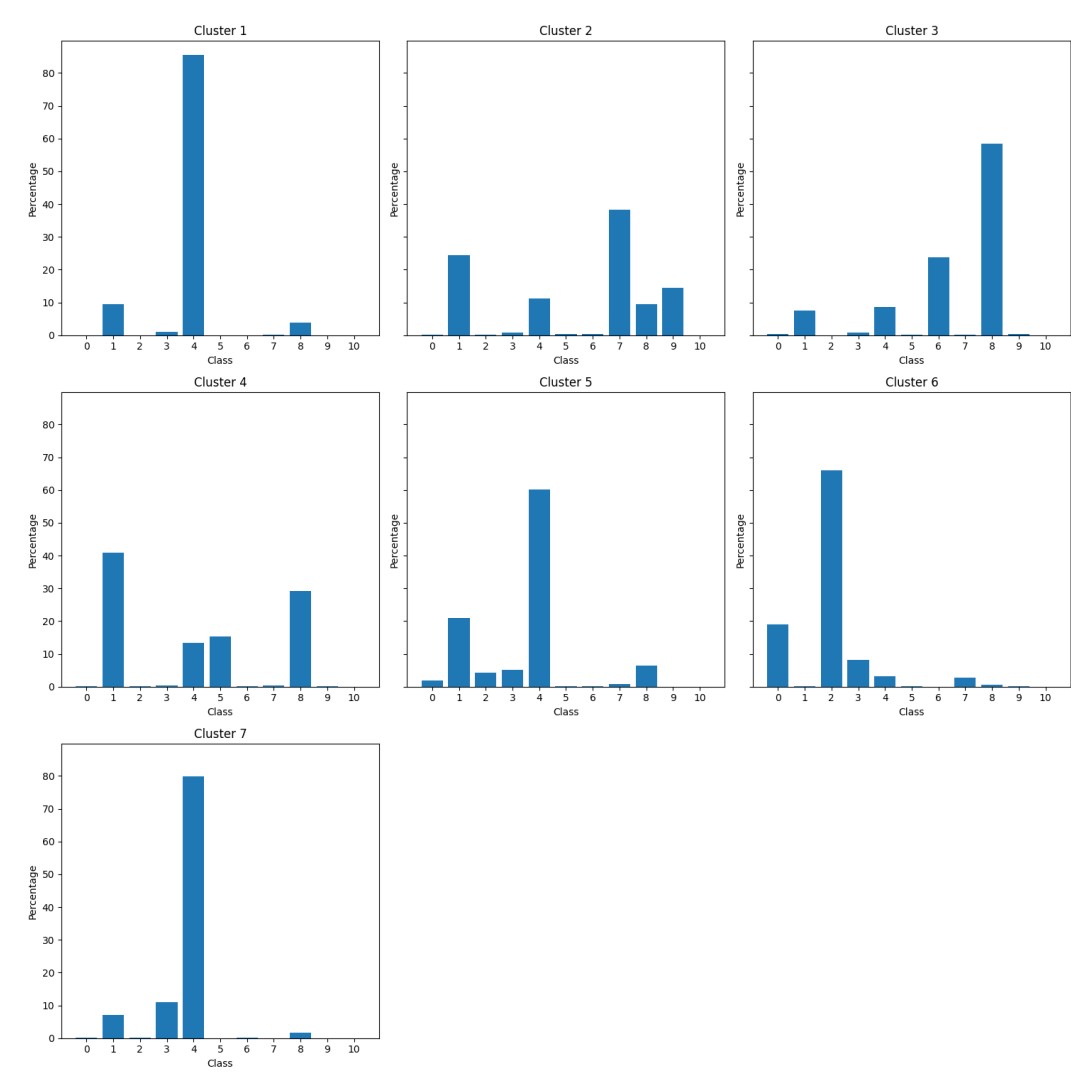

Figure 6: Class distributions across clusters for the Amazon dataset.

clustering captures shared structural or feature-based similarities among nodes, which can serve as a foundation for localized training. However, the presence of mixed-class clusters highlights the need for downstream methods that can handle intra-cluster class diversity effectively.

