# OpenReview forum: "ECGN: A CLUSTER-AWARE APPROACH TO GRAPH NEURAL NETWORKS FOR IMBALANCED CLASSIFICATION."
_ICLR.cc/2025/Conference — Submitted to ICLR 2025_

### Official Review · Reviewer_8tSN · 2024-10-30

**Soundness:** 2
**Presentation:** 3
**Contribution:** 2
**Rating:** 5
**Confidence:** 4

**Summary:**

The paper introduces ECGN, a new method in Graph Neural Networks designed to improve node classification in class-imbalanced datasets by leveraging cluster-specific embeddings and synthetic node generation. ECGN operates by first pre-training subclusters to capture local graph structures, then generating synthetic nodes in boundary areas of minority clusters to enhance classification for underrepresented classes, and finally integrating these local representations into a global structure. This approach outperforms existing models, achieving up to an 11% improvement on benchmark datasets, highlighting its ability to maintain both global and local graph consistency for balanced, high-accuracy node classification.

**Strengths:**

- The paper is well-written, clear, and easy to follow.
- Extensive experiments are conducted to validate ECGN's performance, and the inclusion of the code enhances reproducibility and supports the credibility of results.
- ECGN is highly adaptable, functioning as a versatile enhancement for any GNN backbone, which underscores its potential generalizability across various GNN architectures.

**Weaknesses:**

- Motivation Clarity: Imbalanced classification and node clustering on graphs are two orthogonal problems. The motivation behind combining these two aspects is not sufficiently clear, and further elaboration is necessary to explain why clustering is beneficial for addressing class imbalance.
- Related Work: The related work lacks coverage of recent advancements in class-imbalanced node classification. Key methods such as "GraphENS: Neighbor-Aware Ego Network Synthesis for Class-Imbalanced Node Classification" (ICLR 2022), "TAM: Topology-Aware Margin Loss for Class-Imbalanced Node Classification" (ICML 2022), "GraphSHA: Synthesizing Harder Samples for Class-Imbalanced Node Classification" (KDD 2023), and "Class-Imbalanced Graph Learning without Class Rebalancing" (ICML 2024) should be discussed and ideally included in the empirical comparisons.
- Inappropriate claims: The statement in line 54, “uniform updates can make the GNN overlook rich local structures,” is Inappropriate. Through simple uniform aggregation, message passing in GNNs can also capture local structures.
- Evaluation: The F1-score improvements shown in Table 2 are modest across all datasets. Incorporating additional evaluation metrics for imbalanced learning, such as balanced accuracy would provide a more nuanced view of the model’s performance.

**Questions:**

Please refer to weaknesses above.

---

> ### Author Response · Authors · 2024-11-22
> **Response to Reviewer 8tSN**
>
> We are deeply grateful to the reviewer for their insightful and constructive comments. The manuscript has been thoroughly revised to incorporate all suggested changes, with new updates highlighted in red for clarity. A concise, point-by-point rebuttal has also been provided to explain how each comment has been addressed.
>
> **Motivation for combining imbalanced classification and node clustering:** In many real-world settings, the minority classes lack enough samples (i.e., nodes of these classes). To compensate, we need better embeddings for these few samples. Our cluster-specific GNNs lead to better embeddings by exploiting the cluster structure of real-world graphs. These improved embeddings are less useful for the majority-class since they have many sample nodes. Hence, cluster-specific GNNs and cluster-aware SMOTE become useful primarily in imbalanced classification tasks. We have also clarified this in the manuscript.
>
> **Comparisons against recent papers, and new evaluation metrics:** We have added comparisons against two recent SOTA methods named GraphENS and TAM. Our method consistently equals or outperforms them in F1 score. The results are **statistically significant** at the _p<0.1_ level (Table 1). We have added comparisons based on **balanced accuracy**, and these show the same pattern. We have also reorganized Section 2.2 to discuss these and other recent papers noted by the reviewer.
>
> **Message passing in GNNs can also capture local structures:** We have revised the statement to: _“While standard GNNs use uniform update rules that aggregate information from neighboring nodes, they may not fully capture the rich local patterns and community-specific behaviors present in the graph. By incorporating cluster-aware updates, we aim to enhance the model’s ability to learn these localized structures.”_
>
> We hope the revisions and responses effectively resolve all of your concerns. We respectfully request your reconsideration of the manuscript and look forward to receiving your feedback.

---

> > ### Comment · Reviewer_8tSN · 2024-11-27
> >
> > I would like to thank the authors for their detailed response. However, due to the inadequate literature review and model inefficiency mentioned by other reviewers, I believe the submission is still not ready for publication in ICLR and thus will keep my original score.

---

> > > ### Author Response · Authors · 2024-12-01
> > > **Response to Reviewer 8tSN**
> > >
> > > Thank you for the response. We will try to add more literature review and clarify the time consumption in the final version of the manuscript. We thank the reviewer for reviewing our manuscript.

---

### Official Review · Reviewer_4efw · 2024-11-01

**Soundness:** 2
**Presentation:** 2
**Contribution:** 3
**Rating:** 5
**Confidence:** 4

**Summary:**

This paper introduces ECGN, a framework that enhances Graph Neural Networks (GNNs) by incorporating cluster-specific training and synthetic node generation to address class imbalance in graph data, applicable to node classification tasks. ECGN employs a novel technique called Cluster-Aware SMOTE, which generates synthetic minority-class nodes, improving the decision boundary between classes and enhancing overall model performance. By combining cluster-aware updates with a global integration step, ECGN captures both local cluster-specific information and global graph structures. The model was evaluated on five benchmark datasets, including Citeseer and Reddit, and consistently outperformed existing state-of-the-art methods, with up to an 11% improvement in F1-score on imbalanced node classification tasks.

**Strengths:**

1. This paper proposes a novel framework for addressing the problem of label imbalance in graph data.

2. The method demonstrates strong performance on specific benchmark datasets.

3. The code is publicly available, which is beneficial for the community in terms of reproducibility and further research.

**Weaknesses:**

1. The summary of related work is incomplete. See question 1.

2. The description of the methodology is not clear enough. See questions 2, 3, and 4.

3. The experiments are not comprehensive enough. See questions 5 and 6.

**Questions:**

1. The summary of related work is incomplete. Considering that the main contribution of this paper lies in addressing the graph class imbalance problem, the authors should discuss recent papers that tackle this issue, such as [1][2][3][4][5][6], to strengthen the paper's motivation and contribution.

2. The methodology section lacks clarity. The most confusing aspect is the distinction between class and cluster. Specifically, when clustering algorithms like LSH group nodes with similar features into a cluster, is there any correlation between the clusters and the node labels (classes)? The authors should provide a specific example or diagram illustrating how classes and clusters relate in their approach.

3. Continuing from question 2: To what extent does the quality of the clustering algorithm affect the final results? If the clustering algorithm is optimal and clusters nodes perfectly according to their labels, would that improve prediction performance? How should we interpret the statement on line 200, "The choice of clustering algorithm is orthogonal to our method"? Does this imply that the clustering results have no impact on the final outcome of the method? It is recommended that the authors add an ablation study or sensitivity analysis to quantify how different clustering algorithms or their qualities affect the final results.

4.  Continuing from question 2: On line 257, the paper states, "For each minority node v in class Y_m, we compute its connectivity to nodes in other clusters." This statement blurs the distinction between class and cluster. How should we understand the relationship between class and cluster in this context?

5.  The experiments are insufficient. On one hand, the authors mention that the baselines are somewhat outdated. It is recommended to compare against more recent methods, such as [1][2][3][4][5][6]. On the other hand, even when comparing with the baselines mentioned in the paper, the performance improvements across several datasets are marginal, with numerical increases of only 0.02 to 0.03, representing less than a 5% improvement. It is suggested that the authors conduct comparisons with more updated work; this way, even modest improvements could be considered acceptable in the context of advancing the state of the art.

6.  Continuing from question 5: The authors are encouraged to provide training time and training parameters such as training time per epoch, total training time, and memory usage for both ECGN and baseline methods in the paper or appendix. Given that the framework appears to require training multiple GNNs, it is important to clarify whether the performance improvements come at the cost of significantly higher computational overhead.

References:
[1]	Park, Joonhyung, Jaeyun Song, and Eunho Yang. "Graphens: Neighbor-aware ego network synthesis for class-imbalanced node classification." International conference on learning representations. 2021.
[2]	Qian, Yiyue, et al. "Co-modality graph contrastive learning for imbalanced node classification." Advances in Neural Information Processing Systems 35 (2022): 15862-15874.
[3]	Wang, Yu, Charu Aggarwal, and Tyler Derr. "Distance-wise Prototypical Graph Neural Network for Imbalanced Node Classification." Proceedings of the 17th International Workshop on Mining and Learning with Graphs (MLG). 2022.
[4]	Song, Jaeyun, Joonhyung Park, and Eunho Yang. "TAM: topology-aware margin loss for class-imbalanced node classification." International Conference on Machine Learning. PMLR, 2022.
[5]	Zeng, Liang, et al. "Imgcl: Revisiting graph contrastive learning on imbalanced node classification." Proceedings of the AAAI Conference on Artificial Intelligence. Vol. 37. No. 9. 2023.
[6]	Zhou, Mengting, and Zhiguo Gong. "GraphSR: a data augmentation algorithm for imbalanced node classification." Proceedings of the AAAI Conference on Artificial Intelligence. Vol. 37. No. 4. 2023.

---

> ### Author Response · Authors · 2024-11-22
> **Response to reviewer 4efw**
>
> We genuinely appreciate the reviewer’s thoughtful and detailed feedback. All suggested changes have been carefully implemented in the manuscript, with the updates and revisions clearly marked in red for easy identification. Additionally, we have provided a clear and concise point-by-point rebuttal outlining how each comment has been addressed.
>
> **Comparisons against recent papers:** We have added comparisons against two recent SOTA methods named GraphENS and TAM. Our method consistently equals or outperforms them in F1 score. The results are statistically significant at the _p<0.1_ level (Table 1). We have added comparisons based on balanced accuracy, and these show the same pattern. We have also reorganized Section 2.2 to discuss these and other recent papers noted by the reviewer.
>
> **Differences between the clusters and the node labels (classes):** We analyzed the minority node distribution across clusters for Cora and Citeseer (Appendix A.10), and referenced it in the main text. We find that the minority classes are present across all clusters, with (on average) 8% of nodes in each cluster belonging to the minority class. However, our method does not require the clusters and classes to be highly correlated. The clusters represent interesting network structure, so our cluster-specific GNNs can learn better embeddings. These higher-quality embeddings in turn improve the performance of cluster-aware SMOTE.
>
> **Effect of clustering algorithm on results:** We conducted experiments using three clustering strategies: METIS, Locality Sensitive Hashing (LSH), and random clustering. METIS leverages graph structure, while LSH clusters based on features. METIS performs best but LSH is close. Random clustering is worse, but still achieves reasonable performance. This shows that (a) cluster quality is important, (b) clusters based on network structure are more useful than those based on features, and (b) even with uninformative (random) clusters, our method is robust, as other parts of our algorithm pick up the slack. We have added these results in Appendix A.9.
>
> | Method | CORA | CITESEER |
>
> |---------|---------------|---------------|
>
> | METIS | 0.740 ± 0.03 | 0.650 ± 0.03 |
>
> | LSH | 0.729 ± 0.02 | 0.639 ± 0.02 |
>
> | Random | 0.690 ± 0.05 | 0.610 ± 0.03 |
>
> **Connectivity definition in cluster-aware SMOTE:** We have corrected typos in the paper. Connectivity is based on clusters, not classes.
>
> **Training complexity:** A baseline GNN trained for a fixed number of epochs takes O(E) time, where E is the number of edges. ECGN needs O(E(Ci)) time for each cluster Ci, followed by a global GNN that takes O(E) time. But since the sum of E(Ci) equals E, ECGN still needs O(E) time overall, just like a global GNN. Empirically, ECGN increases wall-clock time by a factor of approximately 1.7 for Cora and Citeseer.
>
> **Parameter complexity:** Since ECGN trains one global model plus one model for each of the K clusters, ECGN has (K+1) times the parameters of a baseline GNN. Due to limited time for rebuttal, we will incorporate exact numbers during the camera ready.
>
> We trust that the revised manuscript and our responses sufficiently address all of your concerns. We kindly request your reconsideration of the manuscript and eagerly await your valuable feedback.

---

> > ### Comment · Reviewer_4efw · 2024-11-25
> >
> > Thank you for your detailed responses.
> >
> > Overall, I believe the paper still has two major concerns.
> > First, the related work considered in the experiments is insufficient. I noticed that Reviewer 8tSN also highlighted this issue. Although the authors have added two comparisons, more recent works, such as the ones I mentioned earlier—"Zhou, Mengting, and Zhiguo Gong. GraphSR: A Data Augmentation Algorithm for Imbalanced Node Classification. Proceedings of the AAAI Conference on Artificial Intelligence, Vol. 37, No. 4, 2023"—and those mentioned by Reviewer 8tSN—"GraphSHA: Synthesizing Harder Samples for Class-Imbalanced Node Classification" (KDD 2023) and "Class-Imbalanced Graph Learning without Class Rebalancing" (ICML 2024)—are still missing. I understand that the authors may not be able to include all the experiments in a short time, but I would prefer to see results for more recent works to evaluate the proposed method more objectively.
> > Second, the issue of time consumption remains a concern. I noticed that Reviewer ntpA also pointed this out. According to the authors' response, the proposed method incurs a 1.7× increase in training time while yielding only marginal improvements across several tasks.
> >
> > I will maintain my score.

---

> > > ### Author Response · Authors · 2024-12-01
> > > **Response to reviewer 4efw**
> > >
> > > Thank you for the response. It is true that due to limited time during rebuttal, we were not able to add more related works in the experiments section. We will try to add more related works and clarify the time consumption even better in the final version of our work. We thank the reviewer for reviewing our manuscript.

---

### Official Review · Reviewer_ntpA · 2024-11-01

**Soundness:** 3
**Presentation:** 2
**Contribution:** 3
**Rating:** 6
**Confidence:** 4

**Summary:**

This paper proposes an improved framework for building graph neural networks for imbalanced node classification tasks. The end-to-end flow includes the following steps:
1. Cluster-Specific Training: The input graph is partitioned into smaller clusters, and graph neural networks are pre-trained for each cluster to generate node embeddings.
2. Cluster-Aware SMOTE: Synthetic nodes are generated for minority classes, with a focus on nodes located near cluster boundaries. The embeddings from Step 1 are used to identify nearest neighbors, enabling interpolation to create new embeddings for the synthesized nodes.
3. Global Integration: All embeddings from Steps 1 and 2 are used to train a global classifier.

The framework leverages local structural information in Step 1 and global information in Step 3, while Step 2 addresses class imbalance. Experimental results demonstrate substantial improvement over previous state-of-the-art, including GraphSMOTE and Cluster-GCN.

**Strengths:**

The overall framework is novel and quite reasonable. It combines the strength from both GraphSMOTE and ClusterGCN while introducing innovative improvements on top of the previous SOTA in various aspects, including:
1. Carrying out SMOTE with a cluster aware approach. Synthesizing nodes around the cluster boundary would be more effective for imbalanced classification problems.
2. Using cluster specific GCN instead of a single GCN at the pre-training stage followed by global aggregation at a later stage. This balanced approach captures both local and global information, with trade-offs in compute and storage efficiency.

The proposed ECGN has been compared to a wide range of methods, and the experimental results consistently outperform the references across all evaluated datasets, demonstrating its effectiveness.

**Weaknesses:**

Presentation:
1. The network architecture for cluster specific pretraining and global aggregation in Figure 1 are illustrative at high level, lacking details of network architecture. It didn’t clarify whether local and global network structures are identical. Additionally, the Global aggregation is only described as “H′ ← convolution of G′ using H′ as node features.” in Algorithm 1, without clear correspondence from Figure 1.
2. The methods in experiment evaluation could be more clearly linked to references or description, specifically “Cluster-Aware SMOTE only” is confusing; see Q1 in the Question section.
3. No information on computational cost has been provided, so it is unclear what trade-offs were made to achieve accuracy improvements.

Reference:
The reference to (Chiang et al., 2019) should include the conference information (KDD ’19) rather than only an arXiv link. Additionally, (Zhao et al., 2021b) is not the correct reference for GraphSMOTE, and the information provided in the reference is also incorrect.

Experiments:
More detailed experiments are expected to clarify the value of Cluster-Aware SMOTE; see Q2 in the Question section.

**Questions:**

Q1:
a) What are the specifics of “Cluster-Aware SMOTE only” in the benchmark results, i.e. Table 2? Section 3.2 described “Cluster-Aware SMOTE” to synthesize nodes, but there is no mention of building a classifier.
b) What’s the difference between “Cluster-Aware SMOTE only” and “ECGN with SMOTE”, and
c) What’s the difference between “Cluster-Aware SMOTE only” and GraphSMOTE?
d) Why is “Cluster-Aware SMOTE only” showing inferior performance than GraphSMOTE if the cluster-aware SMOTE is supposed to be more effective than non-cluster-aware SMOTE?

Q2:
Cluster aware node synthesis in 3.2, especially synthesizing nodes around the cluster boundary seems to be an important and novel step compared to GraphSMOTE. How effective is this approach, what if it reverts to a baseline approach? Is there any experiment validating the contribution of this step?

Q3:
Related to Q2, the embeddings for the synthesized nodes are generated from their neighborhood. Neighborhood nodes could come from different clusters, but all embeddings were pre-trained within each cluster. Was any inter-cluster graph structure lost in this process?

---

> ### Author Response · Authors · 2024-11-22
> **Response to Reviewer ntpA**
>
> We sincerely appreciate the reviewer’s thoughtful and valuable feedback. All suggested revisions have been carefully implemented, and the changes are marked in red within the manuscript for ease of reference. Additionally, we have prepared a brief and consice, point-by-point rebuttal to address each of your comments.
>
> **Cluster-Aware SMOTE only:** This baseline applies Cluster-Aware SMOTE without the full ECGN framework. It involves:
>
> 1.  **Node Selection:** Identify minority nodes at cluster boundaries.
>
> 2.  **Synthetic Generation:** Create synthetic nodes in the feature space using SMOTE.
>
> 3.  **Graph Augmentation:** Add synthetic nodes, inheriting connections from parent nodes.
>
> 4.  **Classifier Training:** Train a standard GNN (e.g., GraphSAGE) on the augmented graph.
>
>
> This approach excludes cluster-specific pre-training and global integration. As shown in Table 2, it improves over standard SMOTE but is outperformed by the complete ECGN framework.
>
> **Differences between Cluster-Aware SMOTE only and ECGN with SMOTE:**
>
> *   **Pre-Training:**
>
>     *   _Cluster-Aware SMOTE only:_ No pre-training.
>
>     *   _ECGN with SMOTE:_ Includes cluster-specific GNN pre-training.
>
> *   **Synthetic Generation:**
>
>     *   _Cluster-Aware SMOTE only:_ Generates in feature space.
>
>     *   _ECGN with SMOTE:_ Generates in latent embedding space.
>
> *   **Global Integration:**
>
>     *   _Cluster-Aware SMOTE only:_ No global integration.
>
>     *   _ECGN with SMOTE:_ Integrates cluster-specific embeddings into a global model.
>
>
> These enhancements in ECGN lead to better synthetic node quality and overall performance.
>
> **Differences between Cluster-Aware SMOTE only and GraphSMOTE:**
>
> *   **Node Selection:**
>
>     *   _Cluster-Aware SMOTE only:_ Based on cluster boundaries.
>
>     *   _GraphSMOTE:_ Random or nearest neighbors without cluster focus.
>
> *   **Operation Space:**
>
>     *   _Cluster-Aware SMOTE only:_ Feature space.
>
>     *   _GraphSMOTE:_ Latent embedding space.
>
> *   **Graph Modification:**
>
>     *   _Cluster-Aware SMOTE only:_ Inherits parent edges.
>
>     *   _GraphSMOTE:_ Adds edges based on embedding similarity.
>
>
> GraphSMOTE better integrates synthetic nodes into the graph structure, enhancing performance.
>
> **Performance Comparison:** _Cluster-Aware SMOTE_ only improves SMOTE by targeting important minority nodes but operates solely in the feature space, missing structural insights. _GraphSMOTE_ leverages the latent space, capturing both feature and structural relationships, resulting in more effective synthetic nodes. _ECGN with SMOTE_ combines cluster-specific pre-training, latent space SMOTE, and global integration, addressing these limitations and outperforming both methods as shown in Table 2.
>
> **Reference Fix:** As per the reviewer's comments, we have fixed all references accordingly.
>
> **Additional Clarifications:**
>
> *   **Summary of Differences:**
>
>     *   _Cluster-Aware SMOTE only:_ Enhances SMOTE with cluster-based node selection in feature space.
>
>     *   _GraphSMOTE:_ Operates in latent space with enhanced graph structure integration.
>
>      *  _ECGN with SMOTE:_ Combines cluster-specific training, latent space generation, and global integration for superior performance.
>
> We trust that the updated manuscript and our responses fully address your concerns. We kindly request your reconsideration of the manuscript and look forward to your feedback.

---

> ### Comment · Reviewer_ntpA · 2024-11-26
>
> Thanks for updating the manuscript and providing detailed responses.
>
> Presentation:
> Overall the presentation is slightly better than the original version with the updates.
> The question about "Cluster-Aware SMOTE only" was explained in the authors' response but it could still be unclear if reading the manuscript directly.
> The question about computational cost wasn't fully answered. The manuscript provided theoretical analysis of computation complexity and algorithm design considerations to reduce computation, but actual training and inference time haven't been shared.
> Update: noticed from the author comment to reviewer 4efw that training time increase has been shared, this type of information should be incorporated into the manuscript.
>
> I'd like to maintain the presentation score as 2.
>
> Soundness and Contribution:
> Good to see two additional approaches have been added for comparison and the proposed method still outperforms them in the benchmarks, just the margin is now smaller.
> I'll keep both the Soundness and Contribution scores as 3.
>
> Overall rating:
> I'd like to keep the overall rating as 6, since I believe there is value from the proposed method.

---

> > ### Author Response · Authors · 2024-12-01
> > **Response to Reviewer ntpA**
> >
> > We thank the reviewer for acknowledging the method. We will try to make the manuscript better with additional experiments, and further justification of the method in the next version.

---

### Official Review · Reviewer_kTym · 2024-11-01

**Soundness:** 2
**Presentation:** 1
**Contribution:** 2
**Rating:** 3
**Confidence:** 4

**Summary:**

This paper introduces ECGN, a method designed to address class-imbalanced node classification. The approach begins by clustering nodes into groups, then computing node embeddings within each cluster through separate GNN models. Using these embeddings, it applies SMOTE to oversample minority nodes, augmenting the original graph, which is then processed by another GNN model for further training. Extensive experiments on datasets of varying sizes are conducted to evaluate the model, including ablation studies to assess the contribution of each component.

**Strengths:**

1. The experiments are thorough, covering datasets of various sizes, including a large-scale dataset.
2. The source code is available for review, and the repository is well-organized.

**Weaknesses:**

1. The paper is poorly organized. It devotes excessive space to related work and includes figures and tables, such as Figure 2, Table 3, and Table 4, that take up significant space without providing much information. Meanwhile, important details are missing from the main paper, particularly about the pre-training step in Proposed Algorithm section, which is crucial as an ablation study references it. However, placing these contents in the appendix forces readers to flip back and forth to follow this paper.

2. I have concerns about the pre-training stage. It appears to involve simple feature aggregation with separate GNNs within each cluster. Are these GNNs trained independently? If so, what is the training objective? If not, why not consider a non-parametric model for feature aggregation instead?

3. The clustering stage raises some concerns. Given the high class imbalance in the graph, minority nodes may end up grouped with a large number of majority nodes within the same cluster, leading to a poor clustering quality. Have the authors considered this possibility?

4. Equations in the paper lack indexing, making them difficult to reference. Additionally, in line 257, the paper states that connectivity measures the connections between minority nodes and nodes in other clusters. However, the equation that follows counts connections between minority nodes and nodes in other classes.

5. Standard deviations are missing from the reported results. This is particularly concerning given that each model is only run four times, which could yield results with high variability.

6. In Table 2, the performance difference between ECGN with and without SMOTE is minimal. This raises questions about the necessity of this component. Is synthesizing minority nodes essential for addressing class imbalance in this context? Again, the absence of standard deviation reporting makes it unclear whether this improvement is statistically significant.

7. In the related work section, the paper critiques previous generative methods, stating, “altering the graph structure introduces complexities, and finding the optimal mixing ratio between node features can be difficult, often leading to noisy results that hurt performance.” However, this work does not address these issues, as its node synthesis method appears to share the same limitations as previous node generative approaches.

**Questions:**

See Weakness.

---

> ### Author Response · Authors · 2024-11-22
> **Response to review of Reviewer kTym**
>
> We sincerely thank the reviewer for their thoughtful and insightful comments. The manuscript has been updated with all new changes and revisions clearly highlighted in red text for ease of review. We have carefully addressed all of your feedback and provided a concise, point-by-point rebuttal detailing the corresponding updates in the manuscript to ensure clarity and efficiency.
>
>
> **New baselines, and statistical significance of results:** We have added comparisons against two recent SOTA methods named GraphENS and TAM. Our method consistently equals or outperforms them in F1 score. The results are statistically significant at the _p<0.1_ level. We have added comparisons based on balanced accuracy, and these show the same pattern. We have updated our results (Table 1) with standard deviations and marked the statistically significant differences.
>
> **Paper organization:** We have substantially modified the manuscript in line with your feedback. In particular, we have condensed the related work section, and moved Figure 2 and Tables 3 and 4 to the appendix. We have added a comprehensive description of the pretraining step (Section 3.2). We have also reorganized much of the text to emphasize points raised by the reviewers. Finally, we have reformatted the equations and fixed all references to them.
>
> **Details of pretraining:** As the reviewer noted, we train separate GNNs independently on each cluster. Each cluster-specific GNN is initialized with the same parameters to ensure consistency across clusters. The training objective for each cluster-specific GNN is the same as that of the global model. Specifically, each GNN is trained to minimize the node classification loss within its respective cluster.
>
> **GNNs versus nonparametric models:** While non-parametric models can perform feature aggregation, they lack the ability to learn complex, non-linear relationships and adaptive representations inherent in GNNs. Our approach leverages the expressive power of GNNs to capture both structural and feature-based information within each cluster.
>
> **Minority-class nodes being overwhelmed in clusters:** We analyzed the minority node distribution across clusters for Cora and Citeseer (Appendix A.10). We find that the minority classes are present across all clusters, with (on average) 8% of nodes in each cluster belonging to the minority class. Thus, the minority class does not get overwhelmed.
>
> **Connectivity metric uses clusters, not classes:** For each minority node, we compute its connectivity to nodes in other _clusters_, not other _classes_. We have fixed the equations to clarify this.
>
> **The performance difference between ECGN with and without SMOTE:** ECGN with SMOTE consistently performs across all datasets, and the improvements are often statistically significant (Table 1). For Citeseer, the F1 score improves by 6.5%.
>
> **Emphasis on using clusters:** Our experimental results show that merely incorporating cluster-based training significantly improves performance across several baseline models, including GraphSAGE. This highlights the versatility and impact of the cluster-aware design. The synthetic node generation improves performance, but that is not the sole contribution of our work.
>
> We hope this concise response addresses and manuscript revisions address all of your concerns comprehensively. We kindly request your reconsideration of the manuscript and look forward to your valuable feedback.

---

> > ### Comment · Reviewer_kTym · 2024-11-25
> > **Response to Authors**
> >
> > Thank you for your detailed response. Below are my comments and observations:
> >
> > 1. Paper Organization: The organization of the paper remains challenging to follow. For instance, Table 1(e) spans two columns, which wastes a lot of space, while other contents are moved entirely to the Appendix. This structure makes it difficult to navigate the paper cohesively. Additionally, Figure 1 could be resized to be more concise without losing clarity.
> >
> > 2. GNNs versus Nonparametric Models: To support your claim, it is necessary to include empirical comparisons with nonparametric models. This would strengthen the justification and provide a more comprehensive perspective on your approach.
> >
> > 3. Minority-Class Nodes in Clusters: A clearer explanation is needed regarding why the minority class does not get overwhelmed when it constitutes only 8% within each cluster. Intuitively, this suggests a significant imbalance, with majority nodes being roughly nine times more prevalent than minority nodes.
> >
> > 4. Performance Difference Between ECGN with and Without SMOTE: Upon reviewing the revised paper, the performance with and without SMOTE appears to be within statistical margins of difference. This suggests that incorporating SMOTE does not yield a significant improvement in performance, warranting further exploration or justification.
> >
> > Overall, while the paper has potential, I believe it is not yet ready for publication. I will maintain my current rating.

---

> > > ### Author Response · Authors · 2024-12-01
> > > **Response to Reviewer kTy**
> > >
> > > Thank you for the response, and acknowledging the potential of our work. We will try to even better reorganize the paper, and provide further claims on nonparametric models as well as minority class node issues in the final version of the manuscript. We thank the reviewer for reviewing our manuscript.

---

### Author Response · Authors · 2024-11-22
**Common Responses to Reviewers**

Common Changes Addressed in the Manuscript
==========================================

We sincerely thank the reviewers for their insightful and constructive feedback. Due to common reviews from the reviewers, we have commonly updated the manuscript by incorporating all new changes, and revisions are clearly highlighted in red at the beginning of the manuscript, and appendix to facilitate easy identification.

Below is a summary of the major updates and revisions made to the manuscript based on the reviewers' feedback:

*   **Added New Baselines, Added Related Works and  New Metrics**:

    *   Included comparisons with two recent state-of-the-art methods, GraphENS and TAM.
    *  Discussed a couple of more recent state-of-the-art works in related works, and clarified how ECGN is different.
    *   Updated results to feature F1 scores, balanced accuracy, and standard deviations.

    *   Highlighted statistically significant differences (p < 0.1) in Table 1.

*   **Improved Paper Organization and Clarity**:

    *   Condensed the related work section for brevity.

    *   Moved Figure 2 and Tables 3 and 4 to the appendix.

    *   Added a detailed explanation of the pre-training step in Section 3.2.

    *   Reorganized text to align with reviewers' suggestions and enhanced equation formatting.

*   **Clarified Pre-training Details**:

    *   Explained that cluster-specific GNNs are trained independently with consistent initialization and node classification objectives.

    *   Described how cluster-specific embeddings enhance consistency and performance.

*   **Conducted Minority Class Analysis**:

    *   Analyzed minority node distribution across clusters for CORA and CITESEER, showing approximately 8% minority nodes per cluster (Appendix A.10).

*   **Examined Effect of Clustering Algorithms**:

    *   Performed experiments using METIS, LSH, and random clustering.

    *   Added results demonstrating METIS's superior performance and included findings in Appendix A.9.

*   **Corrected Connectivity Metric Definition**:

    *   Clarified that connectivity in cluster-aware SMOTE is based on clusters, not classes.

*   **Fixed References and Provided Additional Clarifications**:

    *   Corrected all references as per reviewers' comments.

    *   Addressed specific points raised by reviewers to ensure clarity and completeness.


We hope these updates effectively address all concerns raised by the reviewers. Thank you for your valuable feedback and for helping us improve the manuscript.

---

### Meta-Review · Area_Chair_CYsT · 2024-12-17

**Metareview:**

### Summary
The paper introduces ECGN, a framework to address class-imbalanced node classification. ECGN employs cluster-specific GNN pre-training to generate node embeddings, Cluster-Aware SMOTE to oversample minority nodes near cluster boundaries, and a global GNN for final integration and classification. The method combines local cluster-level structural information with global graph insights. Extensive experiments on benchmark datasets show that ECGN achieves competitive results, outperforming state-of-the-art methods.

### Strengths
- The paper combines clustering-based local pre-training, cluster-aware SMOTE, and global GNN training to tackle class imbalance, which is a novel approach in the graph learning domain.
- The method is evaluated on diverse datasets, with ablation studies assessing the contributions of key components. Results show improvement over baseline methods.
- The availability of well-organized code ensures that the method can be reproduced and facilitates further research in this area.

### Weaknesses
- Unclear Motivation and Methodology: The motivation for combining clustering and SMOTE to handle class imbalance is underexplained, particularly how clusters relate to node classes.
- Ambiguities exist in key notations and descriptions (e.g., distinction between classes and clusters, undefined symbols in equations).
- The reported F1-score improvements over baselines are minor (often under 5%), raising questions about the necessity of the SMOTE component and overall method complexity.
- The paper lacks comparisons with recent methods in class-imbalanced graph learning, such as GraphENS (ICLR 2022) and TAM (ICML 2022), which could contextualize its contributions more clearly.

While the paper proposes an innovative framework for addressing class imbalance in graph data, it suffers from unclear methodology, modest empirical improvements, and missing comparisons with recent work. The limited motivation for combining clustering with SMOTE further weakens its contributions. Improvements in clarity, stronger baselines, and more rigorous experimental analysis are needed for acceptance.

**Additional Comments On Reviewer Discussion:**

The common major concerns raised by the reviewers were:
- Clarity in Methodology and Presentation: Reviewers highlight ambiguities in the distinction between clusters and classes, the role of clustering, and the unclear motivation for combining clustering with SMOTE. Mathematical notations, equations, and components like the pre-training stage are poorly explained, making the methodology difficult to follow.
- Modest Performance Improvements: The performance gains (e.g., F1-score improvements) reported in the experiments are minor, raising doubts about the effectiveness of the Cluster-Aware SMOTE component and the overall framework's necessity. Standard deviations are missing, and statistical significance of results is not validated.
- Incomplete Related Work and Comparisons: The paper does not discuss or compare its method against recent advancements in class-imbalanced graph learning, such as GraphENS, TAM, and other recent methods from ICLR, ICML, and KDD. This omission weakens the positioning of the work relative to the state-of-the-art.

The reviewers tried to address the above points, which however could not satisfy the reviewers. For example, reviewers wanted to see comparisons with more recent baselines, such as those published in KDD 2023 and ICML 2024, but the authors only included some of the old baselines.

---

### Decision · Program_Chairs · 2025-01-22

Reject